The earliest known titanosauriform sauropod dinosaur and the evolution of Brachiosauridae

Mannion Philip D. philipdmannion@gmail.com 1
Allain Ronan rallain@mnhn.fr 2
Moine Olivier 3
1 Department of Earth Science and Engineering, Imperial College London , London , United Kingdom
2 Centre de Recherche sur la Paléobiodiversité et les Paléoenvironnements, Museum National d’Histoire Naturelle , Paris , France
3 Laboratoire de Géographie Physique: Environnements Quaternaires et Actuels, CNRS/Université Paris 1 Panthéon-Sorbonne/UPEC , Paris , France
Anquetin Jérémy
Electronic publication date: 2017 May 2
Publication date: 2017
Volume: 5
Electronic Location ID: e3217
Received 2017 Jan 11; Accepted 2017 Mar 23
Copyright: ©2017 Mannion et al.
Copyright year: 2017
Copyright holder: Mannion et al.
License: This is an open access article distributed under the terms of the Creative Commons Attribution License, which permits unrestricted use, distribution, reproduction and adaptation in any medium and for any purpose provided that it is properly attributed. For attribution, the original author(s), title, publication source (PeerJ) and either DOI or URL of the article must be cited.
License URL: https://creativecommons.org/licenses/by/4.0/

Keywords: Character correlation, Brachiosauridae, Gondwana, Cretaceous, Mesozoic, Late Jurassic, Oxfordian, Ontogeny, Laurasia, Biogeography, France, Sacral fusion

Funding: SYNTHESYS Project This research received support from the SYNTHESYS Project (http://www.synthesys.info/), which is financed by European Community Research Infrastructure Action under the FP7 Integrating Activities Programme. The funders had no role in study design, data collection and analysis, decision to publish, or preparation of the manuscript.

==============================
Brachiosauridae is a clade of titanosauriform sauropod dinosaurs that includes the well-known Late Jurassic taxa Brachiosaurus and Giraffatitan. However, there is disagreement over the brachiosaurid affinities of most other taxa, and little consensus regarding the clade’s composition or inter-relationships. An unnamed partial sauropod skeleton was collected from middle–late Oxfordian (early Late Jurassic) deposits in Damparis, in the Jura department of eastern France, in 1934. Since its brief description in 1943, this specimen has been informally known in the literature as the ‘Damparis sauropod’ and ‘French Bothriospondylus’, and has been considered a brachiosaurid by most authors. If correctly identified, this would make the specimen the earliest known titanosauriform. Coupled with its relatively complete nature and the rarity of Oxfordian sauropod remains in general, this is an important specimen for understanding the early evolution of Titanosauriformes. Full preparation and description of this specimen, known from teeth, vertebrae and most of the appendicular skeleton of a single individual, recognises it as a distinct taxon: Vouivria damparisensis gen. et sp. nov. Phylogenetic analysis of a data matrix comprising 77 taxa (including all putative brachiosaurids) scored for 416 characters recovers a fairly well resolved Brachiosauridae. Vouivria is a basal brachiosaurid, confirming its status as the stratigraphically oldest known titanosauriform. Brachiosauridae consists of a paraphyletic array of Late Jurassic forms, with Europasaurus, Vouivria and Brachiosaurus recovered as successively more nested genera that lie outside of a clade comprising (Giraffatitan + Sonorasaurus) + (Lusotitan + (Cedarosaurus + Venenosaurus)). Abydosaurus forms an unresolved polytomy with the latter five taxa. The Early Cretaceous South American sauropod Padillasaurus was previously regarded as a brachiosaurid, but is here placed within Somphospondyli. A recent study contended that a number of characters used in a previous iteration of this data matrix are ‘biologically related’, and thus should be excluded from phylogenetic analysis. We demonstrate that almost all of these characters show variation between taxa, and implementation of sensitivity analyses, in which these characters are excluded, has no effect on tree topology or resolution. We argue that where there is morphological variation, this should be captured, rather than ignored. Unambiguous brachiosaurid remains are known only from the USA, western Europe and Africa, and the clade spanned the Late Jurassic through to the late Albian/early Cenomanian, with the last known occurrences all from the USA. Regardless of whether their absence from the Cretaceous of Europe, as well as other regions entirely, reflects regional extinctions and genuine absences, or sampling artefacts, brachiosaurids appear to have become globally extinct by the earliest Late Cretaceous.

Introduction

Sauropod dinosaur diversity reached an apparent peak in the Late Jurassic (Mannion et al., 2011; Upchurch et al., 2011), comprised primarily of a wide array of near-globally distributed neosauropod lineages (diplodocoids and macronarians), as well as some non-neosauropod eusauropods (Wilson, 2002; Upchurch, Barrett & Dodson, 2004). However, nearly all of this Late Jurassic diversity comes from deposits assigned to the last two stratigraphic stages of the Jurassic, i.e., the Kimmeridgian–Tithonian (157–145 Ma), with Oxfordian (164–157 Ma) remains extremely rare (Weishampel et al., 2004; Upchurch & Barrett, 2005; Mannion et al., 2011). This might result from genuinely low sauropod diversity in the Oxfordian and/or a sampling bias, but could also pertain to poorly constrained dating of some Middle–Late Jurassic deposits (e.g., in the USA and East Asia; Mannion et al., 2011; Xing et al., 2015).

A partial skeleton of a sauropod was discovered in 1934, during quarrying in Damparis, in the Jura department of eastern France (Dorlodot, 1934; Dreyfuss, 1934; Viret, 1935). In the original publication, Dorlodot (1934) made some brief comparisons with the sauropod genera ‘Bothriospondylus’ and ‘Morosaurus’, rejecting the latter as a possible attribution. The material was subsequently described by Lapparent (1943), who referred it to ‘Bothriospondylus madagascariensis’, otherwise known from remains from the Middle Jurassic of Madagascar (Lydekker, 1895). Since its discovery, the French specimen has been variously known in the literature as ‘Bothriospondylus madagascariensis’ (Lapparent, 1943; McIntosh, 1990; Wilson, 2002), the ‘Damparis dinosaur’ (Buffetaut, 1988; Buffetaut, 1994), the ‘Damparis sauropod’ (Allain, Pereda & Suberbiola, 2003; D’Emic, 2012), and the ‘French Bothriospondylus’ (McIntosh, 1990; Mannion, 2010; D’Emic, 2012; Mannion et al., 2013). Nearly all authors to have commented upon this specimen have expressed doubt on Lapparent’s (1943) referral to ‘Bothriospondylus madagascariensis’ (Buffetaut, Cuny & Le Loeuff, 1991; Buffetaut, 1995; Upchurch, 1995; Salgado & Calvo, 1997). This was reinforced in a recent revision of ‘Bothriospondylus’, which considered the genus to represent a nomen dubium (Mannion, 2010). Furthermore, Mannion (2010) demonstrated that the type of ‘B. madagascariensis’ is based on indeterminate material, and represents an unrelated non-neosauropod eusauropod. The ‘French Bothriospondylus’ has been regarded as a brachiosaurid since its initial description by Lapparent (1943), with this identification based on overall similarities (Lapparent, 1943; McIntosh, 1990; Buffetaut, 1995), synapomorphies recovered in previous phylogenetic analyses (Upchurch, 1995; Wilson, 2002; Mannion, 2010; D’Emic, 2012) and, most recently, supported by its first incorporation into a phylogenetic data matrix (Mannion et al., 2013). A late Oxfordian age has been assigned to the marine deposits that yielded the specimen by some authors (Lapparent, 1943; Buffetaut, 1988; Buffetaut, 1992; Allain, Pereda & Suberbiola, 2003), although others have argued for a middle Oxfordian age (Enay, Contini & Boullier, 1988; Broinet al., 1992; Buffetaut, 1994). Either way, if correctly identified and dated to the middle or late Oxfordian, this would make the specimen the earliest known titanosauriform (Mannion et al., 2013).

Brachiosauridae is a clade of titanosauriform sauropods that spanned the Late Jurassic–Early Cretaceous (D’Emic, 2012; D’Emic, 2013; Mannion et al., 2013; D’Emic, Foreman & Jud, 2016), and includes the well-known taxa Brachiosaurus altithorax (Riggs, 1903a) and Giraffatitan brancai (Janensch, 1914) from the Late Jurassic of the USA and Tanzania, respectively (Taylor, 2009). Abydosaurus mcintoshi, from the late Early Cretaceous of the USA (Chure et al., 2010), is also universally considered as a brachiosaurid (Chure et al., 2010; Ksepka & Norell, 2010; D’Emic, 2012; Mannion et al., 2013; Carballido et al., 2015), and the Late Jurassic Portuguese sauropod Lusotitan atalaiensis is probably an additional member of this clade (Antunes & Mateus, 2003; Mannion et al., 2013; D’Emic, Foreman & Jud, 2016; Mocho, Royo-Torres & Ortega, 2016; though see Carballido et al. (2015) for a more basal position). Carballido et al. (2015) recently described Padillasaurus leivaensis, from the late Early Cretaceous of Colombia, which they recovered within Brachiosauridae; if correctly identified, this would make it the first unambiguous occurrence of a South American brachiosaurid. However, there is disagreement over the brachiosaurid affinities of other taxa, with some putative members placed in the sister clade Somphospondyli (e.g., Sauroposeidon proteles), and others recovered outside of Titanosauriformes (e.g., Aragosaurus ischiaticus, Europasaurus holgeri) or even Neosauropoda (e.g., Atlasaurus imelakei, Lapparentosaurus madagascariensis) by different authors (see D’Emic, 2012; D’Emic, 2013; Mannion et al., 2013; Mocho, Royo-Torres & Ortega, 2014; Royo-Torres et al., 2014; Carballido et al., 2015; D’Emic, Foreman & Jud, 2016; Poropat et al., 2016). The topology of Brachiosauridae also varies between analyses (e.g., compare the positions of Brachiosaurus and Giraffatitan in the trees presented by Mannion et al. (2013), Carballido et al. (2015) and D’Emic, Foreman & Jud (2016)), meaning that there is little consensus as to the composition or inter-relationships of Brachiosauridae (see Fig. 1 for previous hypotheses of placements of putative brachiosaurid taxa).

Figure 1 Simplified cladograms summarizing previous hypotheses of placements of putative brachiosaurid taxa.

The four cladograms are based on: (A) Royo-Torres et al. (2014 (using the dataset of Upchurch, Barrett & Dodson, 2004)); (B) Carballido et al. (2015 (using the complete strict consensus tree presented in their supplementary information)); (C) D’Emic, Foreman & Jud (2016); (D) Poropat et al. (2016 (using the complete strict consensus tree presented in their supplementary information)). Brachiosauridae is highlighted in red in each tree.

Based on its brief description and limited illustration in Dorlodot (1934) and Lapparent (1943), the relatively complete nature of the specimen, its current status as an indeterminate brachiosaurid and the earliest known titanosauriform, and the rarity of Oxfordian sauropod remains in general, the ‘French Bothriospondylus’ represents an important specimen for understanding the early evolution of Titanosauriformes, especially brachiosaurid inter-relationships. As such, here we fully re-describe and illustrate the specimen, incorporate it into a revised phylogenetic data matrix, and re-examine the evolutionary history of Brachiosauridae.

Geographical and Geological Context

Geographical location

The village of Damparis (Jura, Franche-Comté) is located in the eastern part of France, near the city of Dole, almost halfway between Dijon and Besançon (Fig. 2). The discovery of the sauropod dinosaur skeleton was made in a quarry owned by the Solvay group (and since 2016 by the Inovyn company), located at the south-eastern border of the village, on the eastern side of the D322 road heading to Tavaux village (Fig. 2). The quarry extends for approximately 1.2 km along a SW-NE axis, with a maximum width of about 0.6 km, and a depth varying between 50 and 60 m. The locus of the discovery is located perpendicularly 350 m eastward from the D322 road, on the southern side of the path leading down into the quarry. In 1934, after the emptying of the sandy-clayey material of the lens that yielded the fossil remains, a wall was constructed to refill the excavation and to allow for the passage of a small train bringing rocks out of the quarry. This wall, shown in Lapparent (1943: pl. 5, fig. 2), is still present today (Fig. 3), aiding the re-identification of the exact stratigraphic position of the discovery, along with the original photographs and descriptions of Dorlodot  (1934).

Figure 2 Location map of the quarry south-eastward of Damparis village and exact location of the palaeontological locus.

The red cross marks the place where the bones and teeth of the dinosaur were found. To the right of the cross, the path slightly turns left following the curvature of the original cliff visible in Dorlodot (1934: fig. 1) and Lapparent (1943: pl.V, fig. 1). The letter ‘T’ marks the place of the truncation (see Fig. 4). The background map is modified from http://www.geoportail.gouv.fr/ and is freely usable (http://professionnel.ign.fr/enseignement-recherche), as is the inset map of France (http://www.pacha-cartographie.com/fonds-de-carte/).

Figure 3 Wall constructed to allow for the refilling of the excavation and passage of the small train on the overlying step during the excavation (A) and today (B).

The surrounding cliff did not progress since the 1930’s and marks the south-eastern limit of the quarry.

One of us (OM) revisited the quarry in 1999, 2015 and 2016, gathering new lithological observations and interpretations, collecting additional ammonite specimens, and producing a revised stratigraphy. Below is a synthesis of the geology and environment of the Damparis quarry, based on these new data and a critical review of the literature.

Lithological description, fossil content, and environmental context

The fossil assemblage was found in a 12 m thick formation formerly named the calcaires fins (Boullier, Contini & Pernin, 1975; see Table S1), composed of limestone bands alternating with clayey beds. At Damparis, the base of this formation is situated at the bottom of a 2 cm thick red clay bed, which is the first bed encountered in the stratigraphic succession. Locally, this clayey bed overlies an unconformity marked by a low angle truncation of underlying micritic limestone beds (Fig. 4) that are characterised by shallow (∼1 m deep), but plurimetric to pluridecametric oscillations. The upper limit of the calcaires fins is placed at the top of a wine-red breccia unit that locally shifts to a wine-red stromatolithic crusting (Boullier, Contini & Pernin, 1975).

Figure 4 Main unconformity of the sequence between the calcaires fins (above) from the calcaires massifs crayeux bioclastiques (below).

It is marked by a sedimentological truncation of the underlying undulating chalky limestone banks and underlined by a 2 cm-thick red clayey bed, the lowermost one of the sequence (Photo: O Moine). See Fig. 2 for its location in the quarry.

In detail, the limestone beds are 0.1 to 1.5 m thick and yield very few fossils. Naked-eye and thin-section observations reveal that their base may be characterised by a few centimetres of red clay flakes and centimetric-wavelength ripple-marks, overlain by woody debris, quartz grains, other dark crystals, and sometimes by crust-forming filamentous algae. All of these elements progressively disappear upward, shifting to a homogeneous micritic limestone with keyvugs, bird’s eyes structures, sparite-filled sheet cracks, and a few Foraminifera. Moreover, thin-sections produced from the base of the first limestone bed demonstrate that lateral textural changes occur. For example, along the southeastern cliff of the quarry (Fig. 2), the texture is similar to that described for the top of the beds near the dinosaur locus; in contrast, 200 m northeast, the limestone is composed of Foraminifera- and shell-based oncolites in a micritic cement, which itself includes numerous angular quartz grains and some micritic intraclasts with a sparite cement. Positive imprints of desiccation cracks have also been observed on the lower surface of these limestone beds, whereas the upper surface is slightly coloured by the impregnation of underlying clay beds.

Alternating clay beds are 1 to 20 mm thick and present a pink-to-red oxidized-iron derived colour. They are deposited on the smooth upper surface of the underlying limestone beds. These sharp transitions imply a certain degree of hardening of the upper surfaces of the limestone beds. Sieved samples from the basal clay bed reveal the presence of mica flakes, glossy and dull angular quartz grains, and woody debris in the 0.063–1 mm fraction, as well as more rounded quartz in the finer fraction (<0.063 mm). Despite slight variations in thickness and colour, all clay beds have the same characteristics.

The fossil discovery was located in a 50 cm thick lens that laterally pinched out into a clay bed. This lens was approximately 9 m long and at least 5 m wide, corresponding to the spatial dispersion of the bones (Fig. 5). It was contoured by regular gentle slopes (Dorlodot, 1934: figs. 7, 8) (Fig. 6), and its base was smooth (Dorlodot, 1934: p. 574, figs. 8, 9). As the exact locus can no longer be accessed (Fig. 3), only the original descriptions of the lens by Dorlodot (1934) remain. They are of primary importance as no other lens has ever been found or is presently visible in the quarry.

Based on the average thickness of 0.7 m for the limestone bed compressed by the lens (Dorlodot, 1934), the lens was located about 2 m above the basal clay bed that defines the base of the calcaires fins Formation (Fig. 6). Consequently, the fossils were not contemporaneous with the basal clay bed of the formation, as argued by Boullier, Contini & Pernin (1975), and later repeated in Enay, Contini & Boullier (1988) and Buffetaut (1988). Reasons for this mistake likely pertain to the similar thickness of the limestone beds overlying both the basal and fossil-bearing clay beds, as well as the comment by Dorlodot (1934) that the top of the overlying limestone bed was 1.3 m beneath the railway, which at that time settled over waste materials that were subsequently removed (Fig. 6).

The fossil-bearing lens showed two different facies. A 15 cm thick conglomerate of fine grained ‘flooding gravel’ limestome lithoclasts, with a pink-to-red friable matrix of sand, clay and calcium carbonate, formed the base of the lens. A thin-section produced from matrix attached to one of the sauropod bones shows an irregularly coloured micritic cement embedding quartz grains and rounded limestone lithoclasts that contain bird’s eyes structures and Foraminifera. Both types of element show a brownish border derived from the cement, which Dreyfuss (1934) attributed to a high content in organic matter. This unit yielded all of the sauropod dinosaur bones, as well as shells of the gastropod Nerinella, which often indicate brackish environments (Lapparent, 1943). A fine, homogeneous and varicoloured dendritic sediment was also present in the lower 15 cm of the lens (Dorlodot, 1934). As well as plant remains (cycads, undetermined stems and rare ferns), almost all of the teeth of the sauropod were found in this fine sediment, along with six theropod teeth (Dorlodot, 1934) that Lapparent (1943) assigned to ‘Megalosaurus insignis’, although only four of them have been found in the Paris Muséum collections (MNHN.F.1934.6 DAM45–48). These theropod teeth should be regarded as indeterminate representatives of Megalosauridae, likely closely related to Torvosaurus. The upper 35 cm of the lens was in direct continuity with the lower unit. In some places, the sediment here displayed the facies of the upper part of the limestone beds, though with red to grey colours, whereas in other places it was a sandy limestone with yellowish patches (Dorlodot, 1934). This upper unit only yielded ‘lizard-skin-like’ vegetation imprints.

Figure 5 Original excavation plan of Vouivria damparisensis n. gen. n. sp. (modified from Dorlodot, 1934).

Figure 6 The locus of the discovery during the excavation (A) and today (B).

(1) Upper banks of the calcaires massifs crayeux bioclastiques; (2) basal clayey bed separating the calcaires massifs crayeux bioclastiques (below) from the calcaires fins (above); (3) upper limit of the basal submetric limestone bank of the calcaires fins underlining the stone wall that now masks the locus of the discovery; (4) clayey bed that laterally connected with the fossil discovery; (5) southern edge of the depression infilling; (6) location of the locus of the fossil discovery; (7) upper limit of the submetric limestone bank overlying the infilling of the depression; (8) waste material and overlying railway mentioned in the original description, but today both removed (Photo: O Moine).

According to the facies definitions of Cariou et al. (2014), the environment oscillated between a semi-restricted lagoon at the top of the limestone beds and a tidal-flat environment (intertidal to supratidal zones) at the top of the clay beds. Carbonate deposition thus occurred in slightly deepening shallow waters, with more or less hydrodynamic zones, under terrestrial influence. This alternated with terrigenous inputs which took place after brief pauses in sedimentation, in subaerial contexts, followed by short emersion phases. This 12 m thick succession of cyclical shallow facies characterises a very slow transgression during which sedimentation equated to accommodation space. Though diachronic, these facies are very similar to those in which were preserved dinosaur trackways at Loulle, Jura, France (Cariou et al., 2014).

Stratigraphic position and age

A detailed account of the stratigraphic position of the calcaires fins Formation, along with paleogeographical implications on a regional scale, are provided in the Supplementary Information. Here we focus on stratigraphic information directly pertinent to the Damparis quarry. Historically, owing to the absence of either associated or more decisive biostratigraphic markers or radiometric dating, the dinosaur assemblage at Damparis was attributed to the stages Astartien (Dorlodot, 1934) and later Séquanien (Lapparent, 1943), based on its position in the stratigraphic succession. These terms are synonymous and were later revised to designate a facies that in Franche-Comté extends from the upper Oxfordian to the lower Kimmeridgian (Enay, Contini & Boullier, 1988). In their synthesis of ammonite faunas of the Dole region, Boullier, Contini & Pernin (1975) attributed the calcaires fins Formation —and consequently the Damparis fossil assemblage—to the Bifurcatus zone (Table S1). Based on this publication, Buffetaut (1988) attributed the dinosaur assemblage to the upper Oxfordian, whereas Broin et al. (1991) placed it in the lower Kimmeridgian (possibly based on Lapparent, 1967). This led to a short debate in the literature (Buffetaut, 1992; Broin et al., 1992), at the end of which Broin et al. (1992 (based on Enay, Contini & Boullier, 1988: p. 302)) proposed a middle Oxfordian age, at the top of the Transversarium zone (see also Buffetaut, 1994).

At Damparis, the calcaires fins Formation immediately overlies a major truncation in the quarry (Fig. 4). Locally, it represents the Ox5 third order cycle sequence boundary (see Supplemental Information 1), based on the long term regression trend characterising depositional environments throughout all underlying units from the basal calcaires hydrauliques of the quarry (Fig. 7). Above the calcaires fins, the discovery of a specimen of the ammonite Perisphinctes cf. quadratus places the calcaires graveleux inférieurs in the Bifurcatus zone (Boullier, Contini & Pernin, 1975). The positioning of the Ox6 sequence boundary at its upper surface (see Supplemental Information 1) also supports its correlation with the lower part of the Marnes de Besançon Formation, whose deposition began during the Stenocycloides subzone (Fig. 5; Table S2). The Damparis fossils thus belong to the Calcaires de Clerval Formation. The few ammonites yielded by this formation indicate that its deposition ranges from the Rotoides to the Stenocycloides subzones in the region (Table S2). Ammonites are known in both members of the Calcaires de Clerval Formation. However, the basal member, the Calcaires de l’Isle-sur-le-Doubs, differs from the upper member, the Tidalites de Mouchard, by the absence of terrigenous inputs, i.e., quartz and clays (Enay, Contini & Boullier, 1988). As far as this argument is reliable, the presence of about 10% of quartz in the Damparis fossil layer (Dreyfuss, 1934), and in the basal clay bed of the calcaires fins, implies that the fossils might belong to the Tidalites de Mouchard Member and thus date from near the top of the Calcaires de Clerval Formation, i.e., most probably from the Stenocycloides subzone. However, the exact position within the Calcaires de Clerval Formation cannot currently be ascertained. An updated stratigraphic column for the Damparis column is presented in Fig. 8.

Figure 7 Chronostratigraphical position of the Damparis fossils.

Cyclostratigraphy and ammonites zones according to Hardenbol et al. (1998), and chronology based on Cohen et al. (2013). Sources for chronostratigraphical ranges of ammonites are given in Table S2. Dark and light blue dashed lines link maximum flooding surfaces of Tethyan and Boreal domains, respectively. Damparis and regional stratigraphic formations are separated by dashed lines, as their exact position in the ammonite biostratigraphic scheme remain to be established.

Taphonomic Context and Environmental Chronology

The sauropod specimen was distributed over an area of approximately 30 m2 (Fig. 5). Skeletal elements appear to have been recovered from a single stratigraphic horizon, and are consistent in preservation. The overall anatomical sequence has been largely maintained, with most of the presacral vertebrae, pectoral girdle and forelimb elements clustering at one end of the locality, and the sacrum, caudal vertebrae and hindlimb elements at the other end. As such, the overall disposition, consistency in size, and lack of duplication of elements suggests the presence of a single sauropod individual at this locality.

Figure 8 Stratigraphic column of the Damparis quarry.

Following the criteria of Badgley (1986), both the original description and excavation map (Fig. 5) indicate that the bone assemblage had an associated degree of articulation, a clustered spatial distribution, and a mostly articulated and partially bone-bed stratigraphic range, which suggests low reworking. The position of the sauropod teeth and left femur suggests a very short, roughly southward transportation. The absence of the skull, most vertebrae and ribs—very sensitive to water streams (Behrensmeyer, 1975)—supports such a process.

Damage of non-biological origin is relatively limited. Bone surfaces are characteristic of: (1) the first abrasion degree (most of the long bones, a few small ones, and all vertebrae) or no abrasion, which suggests very short or no transport (Fiorillo, 1988); and (2) the first weathering stage, implying a burial within 0 and 3 years of death (Behrensmeyer, 1978). Transverse fracturing of most of the bones and deformation of some of them result mostly from diagenetic compaction, as well as partly from excavation (see Dorlodot, 1934), and possibly from further handling.

Damage of a biological origin is less obvious and, if any, also modest. Spiral fracturing has been noted on most of the appendicular elements, as well as some rib fragments. Based on the likely limited amount of transport and rolling, these fractures might result from trampling, chewing, or breakage by carnivorous taxa. Despite the presence of theropod teeth in the assemblage, no obvious tooth marks have been observed. Nevertheless, tooth marks are not commonly observed in reptilian fossil assemblages (e.g., Buffetaut & Suteethorn, 1989; Fiorillo, 1991). Thus, it remains uncertain whether the lack of the left humerus results from predation or water transport, for example. Trampling was also probably limited, as only one extremity of an ulna was found deeply embedded in the underlying limestone (Dorlodot, 1934).

The initial local context was a semi-restricted lagoonal environment subject to sporadic emersions in a regressive setting, leading to the formation of limestone beds. The deposition of the sauropod body occurred before the limestone surface was completely hardened, which led to the formation of a hollow. The regressive setting enabled the scavenging of the sauropod body by theropods. Such associations of sauropod skeletons and theropod teeth are indeed good markers of emersion (Buffetaut, 1988; Buffetaut & Suteethorn, 1989). This short and limited emersion (several years at most), close to the coast, was sufficient for the limestone surface to harden and to prevent its erosion.

During the subsequent transgression, a prevailing coastal environment of a semi-restricted lagoon, under the influence of brackish continental run-off and involving moderate water streams, led to the rapid burial and disturbance of the sauropod skeleton. Based on the heterogeneous facies of the sediment forming the lens, as well as its complicated colouration (possibly resulting from organic body fluids; Schäfer, 1972: p. 23), burial might even have initiated whilst the body was still decaying and being scavenged. Whether the numerous plant residual imprints associated with the bones were trapped in this hollow by the sauropod carcass, possibly constituting a part of its stomach content, or grew in situ during the emersion phase remains uncertain. With the continuation of the transgression, the locus returned to semi-restricted lagoonal conditions, but with a greater influence from continental inputs. From this point on, only diagenesis and the ultimate excavation of the skeleton would have affected the buried material.

Finally, based on the original description of the fossil-bearing lens at Damparis by Dorlodot (1934: see also figs. 2, 3, 8), the lithology of the upper half of the lens varies spatially between a sandy calcareous facies and a micritic facies, which suggests deposition during minor oscillations between proximal and distal lagoonal settings. The lithological disconnection between the lens and the overlying homogeneous limestone bed suggests a subsequent rapid environmental shift to a semi-restricted lagoonal environment. The transition phase documented by the upper half of the lens cannot be observed lateral to the excavation locus in the thin clay bed either because of absence of deposition or as a result of subsequent erosion associated with the rapid transgressive shift. Contrary to what would be expected based on their thinness, the clay beds of the calcaires fins Formation are thus condensed units that might result from multiple sea-level oscillations during regression-transgression cycles, whose preservation began when transgression exceeded sedimentation. Although Dorlodot (1934) favoured an interpretation in which the sauropod carcass was rafted out to sea, along with attached vegetation and detrital sediment, our results support the environmental conclusion of Buffetaut (1988), i.e., an in situ death and burial of the sauropod on emergent marine sediment.

Nomenclatural acts

The electronic version of this article in Portable Document Format (PDF) will represent a published work according to the International Commission on Zoological Nomenclature (ICZN), and hence the new names contained in the electronic version are effectively published under that Code from the electronic edition alone. This published work and the nomenclatural acts it contains have been registered in ZooBank, the online registration system for the ICZN. The ZooBank LSIDs (Life Science Identifiers) can be resolved and the associated information viewed through any standard web browser by appending the LSID to the prefix http://zoobank.org/. The LSID for this publication is: urn:lsid:zoobank.org:pub:5EB3AF68-A8A2-407D-BF1B-5F856C3B505D. The online version of this work is archived and available from the following digital repositories: PeerJ, PubMed Central and CLOCKSS.

Figure 9 Teeth of Vouivria damparisensis (MNHN.F.1934.6 DAM1–DAM 5).

(A–E) labial views, (F–J) mesial views, (K–O) lingual views; (A, F, K) DAM 1, (B, G, L) DAM 2, (C, H, M) DAM 3, (D, I, N) DAM 4, (E, J, O) DAM 5. Scale bar equals 20 cm.

Figure 10 Middle cervical vertebra of Vouivria damparisensis (MNHN.F.1934.6 DAM 6).

(A) Left lateral view, (B) anterior view, (C) right lateral view; (D) posterior view. Abbreviations: cpol, centropostzygapophyseal lamina; di, diapophysis; cprl, centroprezygapophyseal lamina; espol, expanded spinopostzygapophyseal lamina; no, notch; ns, notch; pa, parapophysis; pn, pneumatic foramen; pocdf, postzygapophyseal centrodiapophyseal fossa; podl, postzygodiapophyseal lamina; poz, postzygapophysis; prz, prezygapophysis; ri, ridge; sdf, spinodiapophyseal fossa; spol, spinopostzygapophyseal lamina; sprl, spinoprezygapophyseal lamina. Scale bar equals 10 cm.

Figure 11 Posterior cervical vertebra of Vouivria damparisensis (MNHN.F.1934.6 DAM 7).

(A) Left lateral view, (B) right lateral view, (C) anterior view; (D) posterior view; (E) dorsal view. Abbreviations: acdl, anterior centrodiapophyseal lamina; cprf, centroprezygapophyseal fossa; cprl, centroprezygapophyseal lamina; nc, neural canal; pa, parapophysis; pcdl, posterior centrodiapophyseal lamina; pn, pneumatic foramen; ri, ridge; tprl, intraprezygapophyseal lamina. Scale bar equals 5 cm.

Figure 12 Anterior dorsal vertebra of Vouivria damparisensis (MNHN.F.1934.6 DAM 8).

(A) Left lateral view, (B) right lateral view, (C) anterior view; (D) ventral view. Scale bar equals 5 cm.

Systematic Paleontology

Sauropoda Marsh, 1878	
Eusauropoda Upchurch, 1995	
Neosauropoda Bonaparte, 1986	
Macronaria Wilson & Sereno, 1998	
Titanosauriformes Salgado, Coria & Calvo, 1997	
Brachiosauridae Riggs, 1904	
Vouivria n. gen. urn:lsid:zoobank.org:act:B06BCF72-56A8-4DD6-BC27-CC0BA6D0092D	
Vouivria damparisensis n. sp. urn:lsid:zoobank.org:act:CCAA960C-6A39-46A4-8AC9-70D8BA816647	
Figs. 9–38	
Bothriospondylus madagascariensisLapparent, 1943	
Damparis dinosaur Buffetaut, 1988	
French “Bothriospondylus madagascariensis” McIntosh, 1990	
Bothriospondylus madagascariensisWilson, 2002	
Damparis sauropod Allain, Pereda & Suberbiola, 2003	
Brachiosauridae indet. Mannion, 2010	
‘French Bothriospondylus’/Damparis sauropod D’Emic, 2012	
‘French Bothriospondylus’ Mannion et al., 2013	

Etymology: The generic name is derived from the old French word ‘vouivre’, itself from the Latin ‘vipera’, meaning ‘viper’. In Franche-Comté, the region in which the holotype was discovered, ‘la vouivre’ (=the wyvern) is a legendary winged reptile. In the homonym novel written by the great French author Marcel Aymé, ‘La Vouivre’ is a beautiful woman who lives in the swamps in the neighbourhood of Dôle (Franche-Comté) and protects a spectacular ruby. The specific name is derived from Damparis, the type locality of the new taxon.

Holotype: MNHN.F.1934.6 DAM 1 to DAM 42 comprising an associated skeleton of a single individual, preserving: five teeth (DAM 1–DAM 5; Fig. 9); a middle–posterior cervical vertebra (DAM 6; Fig. 10); the posterior half of the centrum of a middle–posterior cervical vertebra (DAM42); a posterior cervical vertebra (DAM7; Fig. 11); a middle dorsal vertebra (DAM 8; Fig. 12); a middle–posterior dorsal vertebra (DAM 9; Fig. 13); numerous thoracic ribs (DAM 41); a partial sacrum (DAM 32; Fig. 14); an anterior caudal vertebra (DAM 10; Fig. 15); left (DAM 25) and right (DAM 26) scapulae (Figs. 16 and 17); right coracoid (DAM 27; Fig. 18); right humerus (DAM 28; Fig. 19); right (DAM 29) and left (DAM 30) ulnae (Figs. 20 and 21); right carpal (DAM 19; Fig. 22); proximal half of left metacarpal I (DAM 20; Fig. 23); left metacarpal II (DAM 20; Fig. 24); left metacarpal III (DAM 22; Fig. 25); right metacarpal III (DAM 24); left metacarpal IV (DAM 23; Fig. 26 (see Fig. 27 for reconstructed manus)); left manual phalanx I-1 (DAM 16; Figs. 28A–28C); left manual phalanx II-1 (DAM 15; Figs. 28D–28F); left manual phalanx III-1 (DAM 17; Figs. 28G–28I); left manual phalanx IV-1 (DAM 18; Figs. 28J–28L); two manual (?) ungual phalanges (not located); left ilium (DAM 33; Fig. 29); distal left pubis (DAM 31; Fig. 30); left (DAM 34) and right (DAM 35) ischia (Figs. 31 and 32); right femur (DAM 36; Fig. 33); proximally incomplete left femur (DAM 44); left (DAM 37; Fig. 34) and distal end of right (DAM 38) tibiae; right (DAM 39) and left (DAM 40) fibulae (Fig. 35); left astragalus (DAM 11; Fig. 36); left metatarsal I (DAM 12; Fig. 37); right metatarsal I (DAM 13); distal end of left metatarsal II (DAM 42); right metatarsal III (DAM 14; Fig. 38).

Figure 13 Posterior middle dorsal vertebra of Vouivria damparisensis (MNHN.F.1934.6 DAM 9).

(A) Right lateral view, (B) anterior view, (C) posterior view. Abbreviations: cprl/acpl, centroprezygapophyseal lamina or anterior centroparapophyseal lamina; nc, neural canal; pcdl, posterior centrodiapophyseal lamina; pn, pneumatic foramen; pocdf, posterior centrodiapophyseal fossa. Scale bar equals 5 cm.

Type locality: Solvay group/Inovyn company quarry (47°3′59.63′′N, 5°25′9.01′′E; elevation ∼200 m), Damparis, near Dole, Jura, Franche-Comté, eastern France.

Type horizon and stratigraphic age: Calcaires de Clerval Formation, possibly from the Tidalites de Mouchard Member (Fig. 8). The age of Vouivria damparisensis is constrained to the Rotoides or Stenocycloides subzones, late middle Oxfordian to early late Oxfordian, Late Jurassic (Dorlodot, 1934; Lapparent, 1943; Boullier, Contini & Pernin, 1975; Buffetaut, 1988; Enay, Contini & Boullier, 1988). According to the timescale of Cohen et al. (2013), a range between 163.5 ± 1.0 and 157.3 ± 1.0 Ma can be proposed for the absolute age of Vouivria.

Figure 14 Sacrum of Vouivria damparisensis (MNHN.F.1934.6 DAM 32).

(A) Dorsal view; (B) ventral view. Abbreviations: sv, sacral vertebrae. Scale bar equals 10 cm.

Figure 15 Anterior caudal vertebra of Vouivria damparisensis (MNHN.F.1934.6 DAM 10).

(A) Right lateral view; (B) posterior view; (C) anterior view. Abbreviations: acdl, anterior centrodiapophyseal lamina; cprl, centroprezygapophyseal lamina; di, diapophysis; pcdl, posterior centrodiapophyseal lamina; prdl, prezygodiapophyseal lamina. Scale bar equals 10 cm.

Figure 16 Left scapula of Vouivria damparisensis (MNHN.F.1934.6 DAM 25).

(A) Lateral view; (B) medial view. Abbreviations: ac, acromion; acr, acromial ridge; avp, acromial ventral process; gl, glenoid. Scale bar equals 10 cm.

Diagnosis:Vouivria damparisensis can be diagnosed by four autapomorphies (marked with an asterisk), as well as two local autapomorphies: (1) spinopostzygapophyseal laminae (SPOLs) expand posteriorly close to the spine apex in middle–posterior cervical vertebrae*; (2) well-defined anterior (ACDL) and posterior centrodiapophyseal lamina (PCDL) in anteriormost caudal vertebrae; (3) deltopectoral crest of humerus doubles in mediolateral thickness distally; (4) ventromedial ridge extending distally from the proximal end of metacarpal III bifurcates at approximately one-quarter of the metacarpal length*; (5) ventromedial margin of the proximal third of metacarpal IV forms a flange*; (6) intercondylar ridges between the tibial and fibular condyles, at the distal margin of the posterior surface of the femur*.

Figure 17 Right scapula of Vouivria damparisensis (MNHN.F.1934.6 DAM 26).

(A) Lateral view; (B) medial view. Abbreviations: ac, acromion; acr, acromial ridge; avp, acromial ventral process; gl, glenoid. Scale bar equals 10 cm.

Figure 18 Right coracoid of Vouivria damparisensis (MNHN.F.1934.6 DAM 27).

(A) Lateral view; (B) medial view. Abbreviations: gl, glenoid. Scale bar equals 10 cm.

Figure 19 Right humerus of Vouivria damparisensis (MNHN.F.1934.6 DAM 28).

(A) Anterior view; (B) lateral view; (C) posterior view; (D) proximal view (anterior towards top); (E) distal view (anterior towards top). Abbreviations: co, condyle; dpc, deltopectoral crest; ri, ridge. Scale bar equals 10 cm.

Additional comments

Several elements were incorrectly identified by Lapparent (1943): (1) the cervical vertebra DAM 7 was described as an anterior dorsal vertebra; (2) the dorsal vertebra DAM 9 was considered an anterior caudal vertebra; (3) the right coracoid (DAM 27) was identified as a left element; (4) the left ulna (DAM 30) was identified as the right ulna (DAM 29), and vice versa; (5) the left metacarpal I (DAM 20) was interpreted to be a portion of distal fibula; (6) the left ilium (DAM 33) was described as a right element; (7) the right femur (DAM 36) was regarded as a left element; (8) the fibulae (DAM 39 and DAM 40) were misidentified as radii, even though the right fibula was correctly identified by Dorlodot (1934); and (9) left metatarsals I (DAM 12) and III (DAM 14) were considered right elements.

Figure 20 Right ulna of Vouivria damparisensis (MNHN.F.1934.6 DAM 29).

(A) Anterior view; (B) lateral view; (C) posterior view; (D) medial view; (E) proximal view (anterior margin towards top); (F) distal view (anterior margin towards top). Abbreviations: amp, anteromedial process; alp, anterolateral process; icr, interosseous crest; ol, olecranon. Scale bar equals 10 cm.

Figure 21 Left ulna of Vouivria damparisensis (MNHN.F.1934.6 DAM 30).

(A) Anterior view; (B) lateral view; (C) posterior view; (D) medial view; (E) proximal view (anterior margin towards top). Abbreviations: amp, anteromedial process; alp, anterolateral process; icr, interosseous crest; ol, olecranon. Scale bar equals 10 cm.

Figure 22 Right carpal of Vouivria damparisensis (MNHN.F.1934.6 DAM 19).

(A) Distal view; (B) medial view; (C) proximal view; (D) anterior view. Scale bar equals 5 cm.

The left ischium (DAM 34, which might be the undescribed ‘incomplete left pubis’ mentioned by Lapparent, 1943) and the fragments tentatively identified here as the distal end of a right tibia (DAM 38) and a pubis (DAM 31), were not explicitly mentioned in either Dorlodot (1934) or Lapparent (1943). The following elements were listed and/or figured in Dorlodot (1934) and Lapparent (1943), but could not be located in the MNHN collections: (1) a sixth tooth (described but not figured by Lapparent (1943), and not mentioned by Dorlodot (1934)); (2) an anterior condyle of a cervical vertebra (not figured); (3) a middle dorsal centrum (figured in Lapparent, 1943: pl. II, fig. 3); and (4) two ungual claws (figured in Lapparent, 1943: pl. IV, figs. 13 and 14). Table 1 lists which re-located elements were figured by Lapparent (1943), and Fig. 5 provides a revised quarry map for the holotype skeleton of Vouivria damparisensis, incorporating our new anatomical identifications.

Figure 23 Left metacarpal I of Vouivria damparisensis (MNHN.F.1934.6 DAM 20).

(A) Lateral view; (B) medial view; (C) proximal view (anterior towards top); (D) distal end view (anterior margin towards top). Scale bar equals 5 cm.

Figure 24 Left metacarpal II of Vouivria damparisensis (MNHN.F.1934.6 DAM 21).

(A) Dorsal view; (B) lateral view; (C) ventral view; (D) medial view; (E) proximal view (anterior margin towards top); (F) distal view (anterior margin towards top). Abbreviations: mc I, articular surface for metacarpal I. Scale bar equals 5 cm.

Figure 25 Left metacarpal III of Vouivria damparisensis (MNHN.F.1934.6 DAM 22).

(A) Dorsal view; (B) lateral view; (C), ventral view; (D) medial view; (E) proximal view (anterior margin towards top); (F) distal view (anterior margin towards top). Abbreviations: bp, bulge-like process. Scale bar equals 5 cm.

Table 1 List of re-located elements of Vouivria damparisensis n. gen. n. sp. figured in Lapparent (1943).

Element and specimen number	Plate/figure number	
Tooth (DAM 1)	Plate I, fig. 4	
Tooth (DAM 2)	Plate I, fig. 6	
Tooth (DAM 3)	Plate I, fig. 5	
Tooth (DAM 4)	Plate I, fig. 7	
Tooth (DAM 5)	Plate I, fig. 8	
Cervical vertebra (DAM 6)	Plate II, fig. 4	
Cervical vertebra (DAM 7)	Plate II, fig. 2	
Dorsal vertebra (DAM 8)	Plate III, fig. 4	
Dorsal vertebra (DAM 9)	Plate II, fig. 5	
Sacrum (DAM 32)	Plate III, fig. 1	
Caudal vertebra (DAM 10)	Plate II, fig. 1	
Right humerus (DAM 28)	Plate IV, fig. 1	
Left ulna (DAM 30)	Plate IV, fig. 2	
Right carpal (DAM 19)	Plate IV, fig. 4	
Left metacarpal I (DAM 20)	Plate III, fig. 3	
Left metacarpal II (DAM 21)	Plate IV, fig. 5	
Left metacarpal III (DAM 22)	Plate IV, fig. 6	
Right metacarpal III (DAM 24)	Plate IV, fig. 8	
Left metacarpal IV (DAM 23)	Plate IV, fig. 7	
Left manual phalanx I-1 (DAM 16)	Plate IV, fig. 10	
Left manual phalanx II-1 (DAM 15)	Plate IV, fig. 9	
Left manual phalanx III-1 (DAM 17)	Plate IV, fig. 11	
Left manual phalanx IV-1 (DAM 18)	Plate IV, fig. 12	
Right ischium (DAM 35)	fig. 5	
Right femur (DAM 36)	Plate IV, fig. 15	
Left tibia (DAM 37)	Plate III, fig. 2	
Right fibula (DAM 39)	Plate IV, fig. 3	
Left astragalus (DAM 11)	Plate IV, fig. 16	
Left metatarsal I (DAM 12)	Plate IV, fig. 17	
Left metatarsal III (DAM 14)	Plate IV, fig. 18	

Description and Comparisons

Cranial elements

Teeth

Measurements of the five teeth (DAM 1–5; Fig. 9) are provided in Table 2. DAM 2 is the largest tooth preserved, but is incomplete. DAM 1 comprises a complete crown and root from a slightly smaller tooth. We describe this tooth fully, and then augment this description with information from the other four teeth.

Figure 26 Left metacarpal IV of Vouivria damparisensis (MNHN.F.1934.6 DAM 23).

(A) Dorsal view; (B) lateral view; (C) ventral view; (D) medial view; (E) proximal view (anterior margin towards top); (F) distal view (anterior margin towards top). Abbreviations: fl, medial flange. Scale bar equals 5 cm.

Figure 27 Reconstruction of the metacarpus of Vouivria damparisensis (MNHN.F.1934.6 DAM 20–23) in proximal view.

Scale bar equals 5 cm.

Figure 28 Manual phalanges of Vouivria damparisensis (MNHN.F.1934.6 DAM 15-18).

Left manual phalanx I-1 (DAM16) in (A) dorsal view; (B) proximal view; (C) distal view. Left manual phalanx II-1 (DAM 15) in (D) dorsal view; (E) proximal view; (F) distal view. Left manual phalanx III-1 (DAM 17) in (G) dorsal view; (H) proximal view; (I) distal view. Left manual phalanx IV-1 (DAM 18) in (J) dorsal view; (K) proximal view; (L) distal view. Scale bar equals 5 cm.

Table 2 Measurements of the teeth of Vouivria damparisensis n. gen. n. sp. (MNHN.F.1934.6 DAM 1–5).

An asterisk denotes a measurement based on an incomplete element.

Dimension	DAM 1	DAM 2	DAM 3	DAM 4	DAM 5	
Total apicobasal height of crown and root	81	65*	66*	49*	41*	
Apicobasal height of crown	43	51	44	32	29	
Maximum mesiodistal width of crown	18	19	17	13	13	
Mesiodistal width of crown at base	14	15	14	11	11	
Labiolingual width of crown at base	13	13	12	10	9	
Slenderness Index	2.39	2.68	2.59	2.46	2.23	
Notes.

All measurements are in millimetres.

As in other neosauropods (Upchurch, 1998; Mannion et al., 2013), there is little in the way of mesiodistal narrowing between the crown and the upper portion of the root, although the latter narrows away from the crown. The asymmetrical crown expands gently mesiodistally along its basal two-thirds, before narrowing apically. It is lingually curved, and is also twisted along its axis. In this regard, DAM 1 resembles the maxillary teeth of the brachiosaurids Abydosaurus and Giraffatitan, in which they are twisted axially through an arc of more than 30° (Chure et al., 2010; D’Emic, 2012); however, we cannot determine whether DAM 1 belongs to the upper or lower jaw. The base of the crown has a D-shaped cross section, with a labial surface that is strongly convex mesiodistally, with weakly developed apicobasally oriented labial grooves towards the mesial and distal margins. In contrast, the lingual surface is gently convex mesiodistally along its central portion, forming a low, rounded, apicobasally oriented, midline ridge along the apical third, along with lingual grooves either side of this ridge (better developed on one side than the other). These features of the labial and lingual surfaces are all plesiomorphic for sauropods, but are lost in diplodocoids and titanosaurs (Upchurch, 1995; Upchurch, 1998; Wilson & Sereno, 1998; Mannion et al., 2013). As is the case in most eusauropods (Wilson & Sereno, 1998), apicobasally oriented, anastamosing wrinkles are present throughout the enamel surface of the crown, although both the labial and lingual surfaces become smoother towards the apex (see Holwerda, Pol & Rauhut (2015) for discussion of variation in enamel wrinkling in eusauropod teeth). Carinae are formed along the mesial and distal margins, but do not extend to the base of the crown, differing from the condition in many somphospondylans and rebbachisaurids (Mannion, 2011). Very weakly developed denticles are present on one side of the crown, and are restricted to the apical third. The presence/absence of denticles is relatively ‘plastic’ phylogenetically (Upchurch, 1998), but also even within the dentition of a single individual (Janensch, 1935–1936; Mannion, 2011). The apex of the crown is very slightly worn, forming a low-angled facet (i.e., perpendicular to the long axis of the crown).

Figure 29 Left ilium of Vouivria damparisensis (MNHN.F.1934.6 DAM 33).

(A) Lateral view; (B) medial view. Abbreviations: act, acetabulum; pp, pubic peduncle. Scale bar equals 10 cm.

Figure 30 Distal end of the left pubis of Vouivria damparisensis (MNHN.F.1934.6 DAM 31).

(A) Medial view; (B) distal view. Scale bar equals 10 cm.

Figure 31 Left ischium of Vouivria damparisensis (MNHN.F.1934.6 DAM 34).

(A) Medial view; (B) anterior view; (C) lateral view; (D) proximal view (anterior towards right). Abbreviations: act, acetabulum; fti3, origin site for M. flexor tibialis internus; ip, iliac peduncle; pp, pubic peduncle. Scale bar equals 10 cm.

DAM 2 has more prominently developed lingual grooves and, as was the case in DAM 1, groove depth is asymmetrical. As with DAM 1, denticles are present on one side of the crown, restricted to the apical third. A non-planar wear facet is present at the crown apex of DAM 3, but is restricted to one side of the tooth. As such, it lacks the apical wear facets that characterize the teeth of diplodocoids and many somphospondylans (Wilson & Sereno, 1998; Whitlock, 2011a; Mannion et al., 2013). There are no denticles on the other, unworn side, suggesting that it is the serrated side that has been worn. DAM 4 and 5 are smaller and more asymmetrical than DAM 1–3, and are unworn, with denticles restricted to one side of the apical third; however, extremely subtle serrations are present on the other side in DAM 5.

Figure 32 Right ischium of Vouivria damparisensis (MNHN.F.1934.6 DAM 35).

(A) Lateral view; (B) medial view. Abbreviations: act, acetabulum; fti3, origin site for M. flexor tibialis internus; ip, iliac peduncle; pp, pubic peduncle. Scale bar equals 10 cm.

Figure 33 Right femur of Vouivria damparisensis (MNHN.F.1934.6 DAM 36).

(A) Anterior view; (B) posterior view; (C) distal view. Abbreviations: icr, intercondylar ridges. Scale bar equals 10 cm.

Figure 34 Left tibia of Vouivria damparisensis (MNHN.F.1934.6 DAM 37).

(A) Anterior view; (B) lateral view; (C) posterior view; (D) proximal view (anterior towards top); (E) distal view (anterior towards top). Abbreviations: cc, cnemial crest; lc, lateral condyle; mc, medial condyle; scc, second cnemial crest; tfi, tuberculum fibularis. Scale bar equals 10 cm.

The Slenderness Index (apicobasal length of the tooth crown divided by its maximum mesiodistal width) ranges from 2.23 to 2.68 (Table 2). These values are comparable to those of basal macronarians and several taxa closely related to Neosauropoda, but are much lower than the SI values of diplodocoids and most somphospondylans (Upchurch, 1998; Chure et al., 2010).

Figure 35 Fibulae of Vouivria damparisensis (MNHN.F.1934.6 DAM 39, 40).

(A) Right fibula in anterior view; (B) right fibula in lateral view; (C) right fibula in posterior view; (D) right fibula in medial view; (E) right fibula in proximal view (anterior towards top); (F) right fibula in distal view (anterior towards top); (G) left fibula in anterior view; (H) left fibula in lateral view; (I) left fibula in medial view. Scale bar equals 10 cm.

Figure 36 Left astragalus of Vouivria damparisensis (MNHN.F.1934.6 DAM 11).

(A) Proximal view; (B) distal view; (C) lateral view; (D) posterior view; (E) anterior view. Abbreviations: ap, ascending process; pf, posterior fossa; pp, posterior process; ri, ridge. Scale bar equals 5 cm.

Figure 37 Left metatarsal I of Vouivria damparisensis (MNHN.F.1934.6 DAM 12).

(A) Ventral view; (B) lateral view; (C) proximal view; (D) dorsal view; (E) medial view; (F) distal view. Scale bar equals 5 cm.

Axial skeleton

Cervical vertebrae

Measurements of the axial skeleton are provided in Table 3. Approximate identification of the serial positions of vertebrae was based on comparisons with eusauropods preserving articulated vertebral columns. DAM 6 is a fairly complete middle–posterior cervical vertebra (Cv) (Fig. 10), but it has been mediolaterally compressed, and the right lateral surface of the neural arch and spine is poorly preserved. It is not possible to determine the internal tissue structure. The prominently opisthocoelous centrum has an average Elongation Index (aEI (see Chure et al., 2010)) of 2.15. The ventral surface of the centrum is transversely concave anteriorly (although this has been greatly accentuated by compression), but becomes flat and gradually gently transversely convex posteriorly. As such, it lacks the extensive ventral sulcus that characterizes the postaxial cervical centra of many flagellicaudatans, as well as several additional taxa, including Giraffatitan (Upchurch, 1995; Upchurch, 1998). There is no ventral midline ridge or fossae, and only a weakly developed lateroventral ridge along the posterior half of the centrum, although the latter might have been affected by crushing. The parapophyses have been deflected ventrally via compression, but probably projected primarily laterally, as in most sauropods (D’Emic, 2012). The dorsal surface of each parapophysis is excavated, with this excavation separated from the lateral pneumatic foramen by a horizontal ridge. This is the case in most derived eusauropods, but the cervical parapophyses are unexcavated in most somphospondylans and a small number of other taxa (Upchurch, 1998; Whitlock, 2011a). The excavated parapophyseal surface of DAM 6 is also separated from the anteroventral corner of the lateral surface of the centrum by another ridge. The lateral pneumatic foramen is a deep structure that leaves a thin midline septum, and it extends for most of the centrum length, terminating a short distance from the posterior margin. This pervasive foramen differs from the anteriorly restricted foramina of a number of titanosauriform taxa (Whitlock, 2011a), and the shallow excavations seen in many somphospondylans (Upchurch, 1998; Curry Rogers, 2005). As in most derived eusauropods (Upchurch, 1995; Upchurch, 1998), internal ridges subdivide the foramen (see the right side especially) but, unlike some diplodocoids (Mannion et al., 2012), these internal ridges are not confluent with the lateral surface of the remainder of the centrum. There is no additional foramen on the posteroventral corner of the lateral surface of the centrum, such as that seen in several diplodocine taxa (Whitlock, 2011a). In dorsal view, there is a slight midline notch on the posterior margin of the centrum, although this is not as pronounced as that seen in Europasaurus (Sander et al., 2006) and Giraffatitan (Carballido & Sander, 2014).

Table 3 Measurements of the vertebrae of Vouivria damparisensis n. gen. n. sp. (MNHN.F.1934.6 DAM 6–10 and DAM 32).

Neural arch height was measured from the dorsal surface of the centrum up to the base of the articular surfaces of the postzygapophyses, and neural spine height from this point upwards. Note that the measurements of the widths of the sacral centra are based on Sv1 (measured posterior to sacral ribs) and Sv4.

Dimension	DAM 6 (Cv)	DAM 7 (Cv)	DAM 8 (Dv)	DAM 9 (Dv)	DAM 32 (Sv)	DAM 10 (Ca)	
Centrum length (including condyle)	423	230*	220	∼120	–	–	
Centrum length (excluding condyle)	372	139*	198	∼100	–	88	
Anterior centrum height	–	156	162	192	–	193	
Anterior centrum width	–	213	141	200	∼170	197	
Posterior centrum height	211	149*	173	186	–	182	
Posterior centrum width	135	150*	∼110	199	144	196	
Total length of fused centra	–	–	–	–	730	–	
Neural arch height	76	110*	–	–	–	80	
Neural spine height	215	–	–	–	–	–	
Maximum mediolateral width across sacrum  (including ribs)	–	–	–	–	682	–	
Notes.

All measurements are in millimetres.

* denotes a measurement based on an incomplete element.

The anterior tip of the prezygapophysis is not preserved, but it seems unlikely that much is missing. Even incomplete, it extends beyond the anterior margin of the non-condylar centrum. The centroprezygapophyseal laminae (CPRLs) are too incomplete to determine if they are bifid, as is the case in many diplodocoids (Upchurch, 1995; Whitlock, 2011a; Whitlock, 2011b). The diapophysis is supported by a long, anterodorsally oriented posterior centrodiapophyseal lamina (PCDL), the lateral surface of which is excavated by an elliptical fossa close to the diapophysis. There is also a short anterior centrodiapophyseal lamina (ACDL), and a well-developed postzygodiapophyseal lamina (PODL) projects posteroventrally. The prezygodiapophyseal lamina (PRDL) is present, but has been affected by crushing. The diapophysis has been folded down by compression.

The lateral surface of the neural arch, posterodorsal to the PCDL and ventral to the PODL, is excavated by a postzygapophyseal centrodiapophyseal fossa (POCDF), which is divided into a small ventral and larger dorsal opening. Small foramina appear to be present throughout the fossae. The postzygapophyses do not extend to the posterior margin of the centrum. The interpostzygapophyseal lamina (TPOL) is not preserved, and the centropostzygapophyseal laminae (CPOLs) are partly reconstructed. Unlike many rebbachisaurids (Sereno et al., 2007) and several additional taxa (Wilson & Upchurch, 2009), there is no epipophyseal–prezygapophyseal lamina (EPRL). Epipophyses are also not present, but these are often reduced or absent in posterior cervical vertebrae (Tschopp & Mateus, 2013). It cannot be ascertained whether pre-epipophyses were truly absent. It is also not possible to determine if the lateral surface of the base of the prezygapophyseal process is excavated, as is the case in Giraffatitan and some diplodocoids (Whitlock, 2011b; Tschopp & Mateus, 2013).

The lateral surface of the neural spine, just dorsal to the PRDL and PODL, is excavated by a series of small fossae, which together form the spinodiapophyseal fossa (SDF). These fossae are aligned and curve posterodorsally, with a larger fossa situated further dorsally. A similar series of excavations is present in middle–posterior cervical vertebrae of Giraffatitan (Janensch, 1950), though not Europasaurus (Carballido & Sander, 2014). Other brachiosaurid taxa do not preserve the relevant region of the skeleton, and so we cannot determine how widespread this feature is amongst Brachiosauridae, although there appears to be similar series of fossae in undescribed cervical vertebrae from the Late Jurassic Morrison Formation of Utah, USA (BYU 12866 and 12867; e.g., see Wedel, 2005: fig. 7.2), that have been tentatively referred to Brachiosaurus (Taylor, 2009). The dorsal margin of the SDF, on the lateral surface of the neural spine, is marked by a prominent, horizontal ridge (or thickening) immediately below the spine summit, as is the case in several other basal macronarians (including Giraffatitan) and most diplodocids (Tschopp & Mateus, 2013; Poropat et al., 2016). The anterior margin of the neural spine is very slightly anteriorly deflected; although this is likely to be preservational, it seems that this margin was almost certainly subvertical, rather than posterodorsally inclined. The posterior margin slopes slightly to face posteriorly and partly dorsally. Unlike some diplodocines (Whitlock, 2011a; Tschopp & Mateus, 2013), spinoprezygapophyseal laminae (SPRLs) are not ‘interrupted’, and there is no parallel accessory lamina running posteriorly. Spinopostzygapophyseal laminae (SPOLs) expand posteriorly close to the spine apex. A similar expansion is present in the diplodocine Galeamopus (Tschopp, Mateus & Benson, 2015: fig. 36), but otherwise appears to be unique to Vouivria. The postspinal fossa is infilled by a midline rugosity. Unlike flagellicaudatans, some somphospondylans, and several other taxa (McIntosh, 1990; Upchurch, 1995; Upchurch, 1998; Wilson & Sereno, 1998), the neural spine is non-bifid. It has a small central bulge, as well as equivalent ones at the lateral margins, giving it a sinuous profile in anterior view.

DAM 7 (Fig. 11) is interpreted to be one of the posteriormost cervical vertebrae, rather than an anterior dorsal vertebra as proposed by Lapparent (1943). It preserves the centrum, as well as the lower portion of the neural arch, with the posterior surface incompletely preserved. The ventral surface is transversely concave anteriorly, but flattens and becomes gently transversely convex posteriorly. There are no ridges or fossae along the ventral surface. The anterior articular surface of the centrum forms a strongly convex, prominent condyle, forming a rim around its lateral and ventral margins, and resulting in a strong break of slope where the condyle meets the remainder of the centrum. The posterior articular surface appears to be only gently concave, but this is because its margins are not preserved. Both of the parapophyses can be identified on the very anteroventral corners of the lateral surface of the non-condylar centrum, and the right one is complete enough to show that it is dorsally excavated, with this excavation separated from the lateral pneumatic foramen by a subhorizontal ridge. This parapophyseal placement and morphology is in keeping with a cervical, rather than dorsal, identification for this vertebra. The lateral pneumatic foramen is deep and leaves a thin midline septum. Its posterior margin is tall and vertical, rather than acute as would be expected in an anterior dorsal vertebra of a macronarian (Upchurch, 1998); furthermore, it extends to close to the ventral margin of the centrum, whereas it is usually restricted to the dorsal half in dorsal vertebrae. There is evidence for a dividing ridge at least on the right side. An ACDL and PCDL form the margins to a diapophyseal fossa on the lateral surface of the base of the neural arch. The anterior neural canal opening has a dorsoventrally elongate, oval shape, with the narrowest point of this oval directed dorsally; the posterior neural canal opening is subcircular. There is a midline ridge above the anterior neural canal opening, extending up to the ventral margin of the intraprezygapophyseal lamina (TPRL). This creates a shallow centroprezygapophyseal fossa (CPRF) either side of the midline ridge, each of which is laterally bounded by a CPRL. This morphology is present in a range of eusauropod taxa, including several diplodocoids, Camarasaurus and Giraffatitan (Upchurch & Martin, 2002; Curry Rogers, 2009). It is not possible to determine if the CPRLs are bifurcated because this region is too incomplete.

Dorsal vertebrae

A middle dorsal vertebra (Dv) preserves a near-complete centrum and the base of the neural arch (DAM 8; Fig. 12). Although we cannot be certain, its internal tissue structure does not appear to be camellate. The centrum is taller than wide, although this has been affected by transverse compression. The ventral surface of the centrum is gently convex transversely, and lacks ridges or fossae. As such, it differs from the middle–posterior dorsal centra of the brachiosaurids Brachiosaurus and Giraffatitan in which a midline ventral ridge is present (Upchurch, Barrett & Dodson, 2004). The anterior articular surface of the centrum is convex, but this does not form a prominent condyle. This is the plesiomorphic eusauropod condition in middle–posterior dorsal vertebrae, but strong opisthocoely is retained throughout the dorsal vertebral sequence in macronarians (Salgado, Coria & Calvo, 1997; Wilson & Sereno, 1998), with the exception of Lusotitan (Mannion et al., 2013). The posterior articular surface of DAM 8 is moderately concave. A pneumatic foramen occupies most of the centrum length of the dorsal half of the lateral surface, terminating a short distance from the posterior margin. It is not set within a fossa, contrasting with many somphospondylans (Bonaparte & Coria, 1993; Upchurch, Barrett & Dodson, 2004; Mannion et al., 2013), as well as the brachiosaurid Cedarosaurus (Tidwell, Carpenter & Brooks, 1999; Mannion et al., 2013), and there are no internal ridges dividing the foramen, such as those found in several titanosaurs (Salgado, Coria & Calvo, 1997) and diplodocids (Mannion et al., 2012; Tschopp, Mateus & Benson, 2015). Each lateral pneumatic foramen ramifies deeply, leaving a thin midline septum, as in most derived eusauropods (Upchurch, 1998). There are no laminae on the preserved portion of the neural arch.

DAM 9 (Fig. 13) preserves the deformed centrum and base of the neural arch of a middle–posterior dorsal vertebra (note that it is now missing a small portion of neural arch that was present when figured by Lapparent, 1943: pl. 2, fig. 5). Its ventral surface has been strongly affected by crushing, but clearly lacks ridges or fossae, and was most likely transversely convex. The anterior articular surface of the centrum is convex dorsally, but its ventral half is slightly concave, and it does not form a distinct condyle. This unusual condylar morphology, described as ‘slightly opisthocoelous’ by Tschopp, Mateus & Benson (2015), has also been noted in various diplodocoids, including Apatosaurus, Diplodocus and Rebbachisaurus, the putative diplodocoid Haplocanthosaurus (Wilson & Allain, 2015), and is also present in the posteriormost dorsal vertebra of Brachiosaurus (Riggs, 1904: pl. LXX). The posterior articular surface is gently concave, although it might have been flatter prior to crushing. A moderately deep lateral pneumatic foramen is present (though infilled with matrix), and occupies the dorsal half of the lateral surface of the centrum. It extends from close to the anterior margin of the centrum, to a short distance anterior to the posterior margin. It is not set within a fossa. The anterior neural canal opening is a dorsoventrally tall ellipse, whereas the posterior opening has a semi-circular outline, with a flat ventral margin. On the anterior margin of the lateral surface of the arch, there is a sharp ridge that is likely to be the lateral margin of the CPRL/anterior centroparapophyseal lamina (ACPL). A PCDL can also be recognized and forms the anterior margin of a moderately deep POCDF.

A middle–posterior dorsal centrum figured by Lapparent (1943: pl. II, fig. 3) cannot be located in the MNHN collections. It has deep lateral pneumatic foramina, leaving a thin midline septum, and appears to have lacked a prominent anterior condyle.

Thoracic ribs

Numerous portions of thoracic ribs are preserved, mainly comprising shafts, but some preserve the lower portions of the rib head (DAM 41). No rib heads are complete enough to determine whether the posterior surface was excavated, as is the case in titanosauriforms (Wilson & Sereno, 1998; Wilson & Upchurch, 2009) and some diplodocids (Mannion et al., 2012). The upper portions of some of the rib shafts have a “T”-shaped cross section, and several pieces preserve plank-like shafts, indicative of titanosauriform affinities (Wilson & Sereno, 1998), although other portions have narrower shafts. One rib is missing most of the rib head, but is otherwise complete, measuring 1,430 mm in length, with a plank-like shaft. It was not possible to locate the element illustrated by Lapparent (1943: fig. 4).

Sacral vertebrae

The preserved sacrum (DAM 32) is slightly reconstructed in places, but comprises four fused sacral centra, the bases of the ribs of sacral vertebra 1 (Sv1), and three pairs of sacral ribs (Sv2–4) (Fig. 14). The sacrum has been dorsoventrally compressed, and the dorsal surfaces of the final two sacral centra are incomplete. The four preserved vertebrae presumably represent Sv1–4, with the transversely widest centrum representing Sv1. Sv5 is inferred to be missing, rather than considering that Vouivria only had four sacral vertebrae, because the most posteriorly preserved sacral centrum does not seem to expand transversely enough relative to the dimensions of the anteriormost caudal vertebra. Our interpretation of which end of the sacrum is anterior is based on comparisons with eusauropod taxa preserving five sacral vertebrae (e.g., Osborn, 1898; Hatcher, 1903; Ostrom & McIntosh, 1966). We do not regard our interpretation as pertinent to discussions of sacral vertebral homology (e.g., see McIntosh et al., 1996; Wilson & Sereno, 1998; Carballido et al., 2011a; Carballido et al., 2015; Pol, Garrido & Cerda, 2011), as there is no clear evidence that ontogeny of sacral fusion recapitulates the phylogenetic history of sacral addition (Wilson, 2011). However, it is pertinent to discussions of the order of sacral fusion. The centra of Sv2–4 appear to fuse first in sauropods, and it has been noted that Sv5 tends to co-ossify with Sv4 before Sv1 fuses with Sv2 (e.g., Wilson & Sereno, 1998; Ikejiri, Tidwell & Trexler, 2005). As such, this might suggest that our interpretation of the sacrum of Vouivria is either incorrect, or that this taxon was unusual. However, there are several neosauropod specimens in which the centrum of Sv1 is fused to those of Sv2–4, whereas Sv5 is ‘free’ (see Royo-Torres, 2009), including specimens attributed to Apatosaurus, Camarasaurus and Diplodocus (Williston, 1898; Riggs, 1903b; Wedel & Taylor, 2013), as well as Tastavinsaurus (Canudo, Royo-Torres & Cuenca-Bescós, 2008). Melstrom et al. (2016) described a juvenile specimen of Barosaurus in which the centrum of Sv1 is partially fused to that of Sv2, whereas the centra of Sv4 and Sv5 are not in contact. A juvenile specimen assigned to Giraffatitan also has Sv1 fused to Sv2–3 (Sv4 is anteriorly incomplete), with no Sv5 present (Janensch, 1950). In contrast, the only complete sacrum of Europasaurus has a free Sv1 and fused Sv2–5 (Carballido & Sander, 2014). A second specimen of Europasaurus preserves four fused sacral centra, which were interpreted as Sv2–5 by Carballido & Sander (2014). Although it might be more parsimonious to assume that different individuals of the same species underwent the same sequence of sacral fusion, this does not always appear to be the case (e.g., see examples in Wedel & Taylor (2013): table 1); furthermore, the presence of a convexity on the anteriormost preserved centrum of this second Europasaurus sacrum leads us to speculate that this might in fact represent Sv1. Ultimately, we will need a larger sample of ontogenetic stages for individual taxa before we can develop a greater understanding of sacral fusion in sauropods (Wilson, 2011; Melstrom et al., 2016), but there is no evidence to suggest that the condition in Vouivria is unusual.

The ratio of the mediolateral width across the combined sacral vertebrae and ribs to the average anteroposterior length of a sacral centrum is 3.7. This is lower than the values of diplodocoids and most titanosauriforms, which tend to have ratios greater than 4.0 and 5.0, respectively (Upchurch, 1998). However, the ratio in DAM 32 is comparable to that of the basal macronarians Camarasaurus, Europasaurus and Galveosaurus (Poropat et al., 2016).

The ventral surfaces of the centra are gently convex transversely, lacking ridges or fossae. The exposed anterior and posterior articular surfaces of the sacral centra are fairly flat. A moderately deep pneumatic foramen excavates the lateral surface of the centra of Sv1–2, and probably Sv3 too. The presence or absence of foramina within sacral centra appears to be fairly ‘plastic’ among sauropods (e.g., Upchurch, 1998). It is not possible to determine the internal tissue structure of the sacral vertebrae.

The first fully preserved sacral rib pair emanates from the anterior end of Sv2, projecting posterolaterally, but there is a significant contribution from the posterior end of Sv1 too. A similar pattern of dual vertebral contribution to the second pair of sacral ribs seems to be present in Brachiosaurus (Riggs, 1904: pl. LXXIII, fig. 2) and Giraffatitan (Janensch, 1950: fig. 76), but this does not appear to be the case in taxa such as Omeisaurus (He, Li & Cai, 1988: fig. 32) or Haplocanthosaurus (Hatcher, 1903: pl. V, fig. 1), and is clearly absent in other neosauropods such as Apatosaurus (Gilmore, 1936: fig. 7), Camarasaurus (Osborn, 1904: fig. 2) and Diplodocus (Osborn, 1904: fig. 3).

The second preserved sacral rib pair emanates from the anterior end of Sv3, with a small contribution from Sv2. These sacral ribs project posterolaterally from Sv3, but the posterior deflection is not as developed as in the preceding pair. The third preserved sacral rib pair emanates from the anterior end of Sv4, with a small contribution from Sv3; they project almost entirely laterally. The complete sacral ribs are fused distally to form a sacricostal yoke, as in other derived eusauropods (Wilson & Sereno, 1998). Each sacral rib is narrow along its shaft and anteroposteriorly expanded at its medial and, especially, lateral ends. None of them are perforated by a foramen, as occurs in some titanosaurs (Curry Rogers, 2005), and they all lack ridges.

Caudal vertebrae

DAM 10 (Fig. 15) is one of the anteriormost caudal vertebrae (Cd), possibly Cd1 or Cd2, and preserves the centrum, the lower portion of the arch and the bases of the diapophyses (=caudal ribs). The centrum is anteroposteriorly short, with an aEI of 0.45. This is at the low end spectrum of values, although there is a large amount of variation across Eusauropoda (Upchurch, 1998; Mannion et al., 2013). The internal tissue structure of DAM 10 is fine and spongey, lacking camellae, contrasting with the pneumatised anterior caudal vertebrae of many titanosaurs (Wilson, 2002; Mannion et al., 2013). The anterior articular surface of the centrum is gently concave, although this surface is irregular and becomes flatter towards the margins. In contrast, the posterior articular surface of the centrum is flat. As such, DAM 10 lacks the condition seen in several rebbachisaurids (Carballido et al., 2012) and somphospondylans (González Riga, Previtera & Pirrone, 2009; D’Emic et al., 2013), whereby the concavity of the posterior surface of the centrum is deeper than that of the anterior surface. The ventral surface is transversely convex, curving smoothly into the lateral surfaces. There are no ventrolateral ridges, and chevron facets are absent, further supporting the view that this is one of the first caudal vertebrae. There is no pneumatic fossa on the lateral surface of the centrum, contrasting with the condition in diplodocids (Whitlock, 2011a), some rebbachisaurids (Carballido et al., 2012) and titanosaurs (Mannion et al., 2013; Poropat et al., 2016), and several brachiosaurids (D’Emic, 2012), although a few small vascular foramina pierce this surface.

Although only the bases of the caudal ribs are preserved, they show that they were supported from below by a sharp ACDL and PCDL, as well as a PRDL, and that there was no PODL present. Whereas the presence of a prominent PRDL is fairly common within the anteriormost caudal vertebrae of eusauropods (Chure et al., 2010; Mannion et al., 2013), a well-defined ACDL and PCDL are generally restricted to diplodocoids (Wilson, 2002; Whitlock, 2011a), although Giraffatitan also has an ACDL (Janensch, 1950). A PODL appears to be restricted to diplodocines and some rebbachisaurids (Mannion, Upchurch & Hutt, 2011). Each caudal rib extends from the upper third of the centrum and onto the base of the neural arch, contrasting with their dorsal restriction in rebbachisaurids (Mannion, Upchurch & Hutt, 2011). They are too incomplete to determine if a tubercle was present on their dorsal surfaces, such as that found in a range of eusauropod taxa (D’Emic et al., 2013; Poropat et al., 2016).

The neural canal is large at both its anterior and posterior openings, and has a dorsoventrally tall elliptical shape. A few small vascular foramina pierce the neural canal floor. Unlike diplodocids and some rebbachisaurids (Mannion, Upchurch & Hutt, 2011), no centroprezygapophyseal fossa (CPRF) is present between the dorsal margin of the anterior neural canal opening and the TPRL. Only the bases of the prezygapophyses are preserved, which are supported ventrally by sharp CPRLs. Both a prezygapophyseal centrodiapophyseal fossa (PRCDF) and POCDF are present as shallow excavations. The postzygapophyses have been slightly displaced and are deformed. Although this area is damaged, it appears that a hyposphene was probably present, and that this was likely a prominent structure. The presence of a hyposphene in anteriormost caudal vertebrae is the plesiomorphic sauropod condition, but is lost in many somphospondylans (Upchurch, 1998; Mannion et al., 2013) and rebbachisaurids (Mannion et al., 2012). A prespinal and postspinal fossa is present, but there are no distinct prespinal or postspinal ridges along the preserved base of the neural spine, such as those seen in titanosaurs, Giraffatitan, and most diplodocoids (Mannion et al., 2013). There is no spinodiapophyseal fossa (SDF) on the lateral surface of the lower portion of the neural spine, contrasting with several titanosaurs (Wilson, 2002).

Appendicular skeleton

Scapula

The scapulocoracoid is described with the long axis of the scapular blade held horizontally. Both scapulae are preserved (Figs. 16 and 17); although complete in terms of length (Table 4), both are missing material from the dorsal and ventral margins, with the right scapula (DAM 26) slightly more complete than the left element (DAM 25). The glenoid faces anteroventrally, lacking the distinct medial beveling that characterizes somphospondylans (Wilson & Sereno, 1998). The acromial ridge is anteroposteriorly wide, but quite low. It is oriented approximately perpendicular to the long axis of the scapular blade, although its anterior margin is concave, forming the posteroventral margin of the excavation of the lateral surface of the acromion. There is no distinct excavated area posterior to the acromial ridge, in contrast to several neosauropod taxa (Upchurch, Barrett & Dodson, 2004). There is a subtriangular process on the ventral margin towards the posterior end of the acromion, but it is unclear whether there was a second ventral process on the anterior portion of the scapular blade. As such, Vouivria shares the presence of at least the anterior ventral process with a wide array of titanosauriforms, including Giraffatitan, as well as some basal eusauropods and diplodocids (Bonaparte, González Riga & Apesteguía, 2006; Carballido et al., 2011b; D’Emic, Wilson & Williamson, 2011; Mannion et al., 2013; Tschopp, Mateus & Benson, 2015).

Table 4 Measurements of the pectoral girdle elements of Vouivria damparisensis n. gen. n. sp. (MNHN.F.1934.6 DAM 25–27).

Measurements of scapulae presented as left (DAM 25) then right element (DAM 26), and were taken with the long axis of the scapular blade held horizontally.

Element	Dimension	Measurement	
Scapulae	Anteroposterior length	1,333/1,335	
	Acromion anteroposterior length	427/428	
	Acromion maximum dorsoventral height	675/660*	
	Scapular blade minimum dorsoventral height	174/175	
	Scapular blade maximum dorsoventral height	294*/218*	
Right coracoid	Anteroposterior length	300	
	Dorsoventral height	415	
Notes.

All measurements are in millimetres.

* denotes a measurement based on an incomplete element.

As in most eusauropods, with the exception of many somphospondylans (Wilson, 2002), the base of the scapular blade has a D-shaped cross section, with a gently dorsoventrally convex lateral surface and a flat medial surface. The dorsal margin of this D-shape is transversely thicker than the ventral margin. There is no dorsal or ventral ridge on the medial surface of the proximal portion of the scapular blade, contrasting with several derived titanosaurs (Sanz et al., 1999). Distally, the blade is expanded dorsoventrally.

Coracoid

The right coracoid (DAM 27; Fig. 18) is in two pieces, with a small amount of material missing in between (see Table 4 for measurements). The larger element preserves the ventral two-thirds of the coracoid, whereas the smaller element preserves the anterior two-thirds of the dorsal third of the coracoid; thus, the posterodorsal corner of the coracoid is not preserved, including the coracoid foramen.

In articulation, the dorsal margin of the coracoid lies below the level of the scapular acromion plate, and is separated from the latter by a V-shaped notch, as is the case in all non-titanosaurian sauropods (Upchurch, 1995; Upchurch, 1998). The lateral surface of the coracoid is gently convex along the lower two-thirds and mildly concave dorsally, whereas the medial surface is predominantly flat. Both surfaces become concave towards their mediolaterally thickened anterior, posterior and ventral margins. In contrast to some derived titanosaurs (Upchurch, 1998), the anterodorsal corner of the coracoid is rounded, and there are no tubercles or muscle scars on the lateral surface. The glenoid extends prominently onto the lateral surface of the coracoid; with the exception of some derived titanosaurs, this is a feature that characterizes Neosauropoda and closely related taxa (Poropat et al., 2016). There is no notable ventral notch or infraglenoid lip anterior to the glenoid. These features are present in many eusauropods, but are absent in most diplodocoids and brachiosaurids (Wilson, 2002; Carballido et al., 2012; Mannion et al., 2013).

Humerus

The right humerus (DAM 28; Fig. 19) is complete, but is now in two pieces (see Table 5 for measurements). The humerus to femur length ratio is 0.91. In most sauropods, this value is usually approximately 0.8 or less (Wilson, 2002; Carballido et al., 2012; Royo-Torres et al., 2014; Poropat et al., 2016), but the ratio is close to 1.0 in brachiosaurids (Brachiosaurus, Cedarosaurus and Giraffatitan) and Atlasaurus (Monbaron, Russell & Taquet, 1999; Wilson, 2002; Mannion et al., 2013). As such, Vouivria seems to be closer to the brachiosaurid condition, but does not show as high a ratio as those taxa. The forelimb (humerus and ulna) to hindlimb (femur and tibia) length ratio is 0.97, which is considerably higher than other sauropods for which this ratio can be determined (usually less than 0.8; Upchurch, 1995), with the exception of Atlasaurus (1.06 (Monbaron, Russell & Taquet, 1999)), Cedarosaurus (0.96 (Tidwell, Carpenter & Brooks, 1999)) and Giraffatitan (1.10 (Janensch, 1961)).

Table 5 Measurements of the upper forelimb elements (humerus, ulnae and carpal) of Vouivria damparisensis n. gen. n. sp. (MNHN.F.1934.6 DAM 19, 28–30).

Lengths of the anteromedial and anterolateral proximal arms of the ulna follow the protocol proposed by Upchurch, Mannion & Taylor (2015). An asterisk denotes a measurement based on an incomplete or distorted element.

Element	Dimension	Measurement	
Right humerus	Proximodistal length	1,330	
	Proximal end maximum mediolateral width	415	
	Distance from proximal end to distal tip of deltopectoral crest	630	
	Midshaft mediolateral width	161	
	Midshaft anteroposterior length	93	
	Midshaft minimum circumference	450	
	Distal end mediolateral width	355	
	Distal end maximum anteroposterior length	140	
Left ulna	Proximodistal length	910	
	Proximal end maximum mediolateral width	321	
	Proximal end maximum anteroposterior length	247	
	Anteromedial arm length	229	
	Anterolateral arm length	170	
	Distal end maximum mediolateral width	134	
	Distal end maximum anteroposterior length	123*	
Right ulna	Proximodistal length	915	
Right carpal	Maximum proximodistal height	45	
	Maximum mediolateral width	89	
	Maximum anteroposterior length	78	
Notes.

All measurements are in millimetres.

DAM 28 is a gracile element, with low proximal (0.31), midshaft (0.12) and distal end (0.27) mediolateral width to humerus length ratios, and an average value (Robustness Index of Wilson & Upchurch, 2003) of 0.23. These values are notably lower than those of most sauropods (Curry Rogers, 2005; Carballido et al., 2012; Mannion et al., 2013), but are comparable to those of brachiosaurids (Wilson & Upchurch, 2003; Taylor, 2009; Mannion et al., 2013). However, there are a number of non-brachiosaurid taxa that also have gracile humeri (D’Emic, 2012; Mannion et al., 2013), including the somphospondylan Ligabuesaurus (Bonaparte, González Riga & Apesteguía, 2006).

In anterior view, the proximal margin is convex, and it lacks a prominently developed process for the attachment site of M. supracoracoideus. In these regards, the humerus differs from those of many derived titanosaurs (Upchurch, 1998; González Riga, Previtera & Pirrone, 2009). The proximolateral corner of the humerus is rounded, contrasting with the ‘squared’ proximal ends of the humeri of most somphospondylans (Upchurch, 1999; Wilson, 2002), as well as some basal macronarians (D’Emic, 2012; Upchurch, Mannion & Taylor, 2015) and rebbachisaurids (Carballido et al., 2012). There is little in the way of lateral expansion of the proximal end of the humerus, giving it an asymmetrical outline in anterior view. This is the condition in most titanosauriforms, and contrasts with the ‘hourglass’ outline seen in other sauropod humeri (Tschopp, Mateus & Benson, 2015; Poropat et al., 2016).

The rugose proximal articular surface extends slightly onto the anterior surface of the humerus. However, the posterior surface of the proximal end is damaged, meaning that we cannot determine whether the humeral head formed a prominent posterior bulge, such as that seen in several macronarian taxa, including Giraffatitan (D’Emic, 2012) and Ligabuesaurus (Bonaparte, González Riga & Apesteguía, 2006). As in all neosauropods (Poropat et al., 2016), there is a low striated area for attachment of M. coracobrachialis on the anterior surface of the proximal third; this is situated approximately 200 mm from the proximal end. There is no ridge along the posterior surface of the lateral margin of the proximal third, nor are there bulges for attachment sites for M. scapulohumeralis anterior or M. latissimus dorsi, contrasting with several titanosaurian taxa (Otero, 2010; D’Emic, 2012; Upchurch, Mannion & Taylor, 2015). The posterior surface of the proximal third is transversely convex.

The deltopectoral crest projects anteromedially. Although it is not as medially directed as the deltopectoral crests of many titanosaurs (Mannion et al., 2013), its orientation is similar to that of taxa such as Giraffatitan (see Janensch, 1961: Beilage A). The deltopectoral crest increases in anteroposterior prominence and more than doubles in mediolateral thickness along its distal half, before fading out distally. This distal thickening was originally regarded as a feature restricted to derived titanosaurs (Wilson, 2002), but also characterizes turiasaurs (e.g., Zby (Mateus, Mannion & Upchurch, 2014)). We regard this thickening to be a local autapomorphy of Vouivria. The lateral margin of the shaft is gently concave throughout most of its length, with a fairly straight section restricted to the midshaft. As such, the humerus of Vouivria seems to be closer to the plesiomorphic eusauropod condition of having a concave lateral margin (Mannion et al., 2013), rather than the straight margin that extends along the middle third of the humerus in many taxa (e.g., Giraffatitan; Janensch, 1961), although there is a great deal of variation within Neosauropoda (Curry Rogers, 2005; Mannion et al., 2013).

There is a flange-like ridge on the anteromedial margin, a short distance from the distal end, but it is difficult to ascertain how genuine a feature this might be, as both the medial and lateral surfaces have undergone some deformation in this region. Two well-developed and widely spaced condyles are present on the lateral two-thirds of the anterior surface of the distal end. As such, it retains the plesiomorphic condition, whereas there is a single, undivided condyle in derived somphospondylans (D’Emic, 2012). There is a small ridge on the lateral margin of the anterior surface of the distal end, which would potentially be unusual, but this is likely to be the product of crushing of the humerus. The anterior surface of the medial third of the distal end is flat and featureless. The posterior surface of the distal end is poorly preserved, but the supracondylar fossa is clearly very shallow, and the ridges either side are broad and rounded, rather than sharp. In this regard, it differs from the deep supracondylar fossa that characterizes most derived somphospondylans (Upchurch, Barrett & Dodson, 2004; Mannion & Calvo, 2011; Upchurch, Mannion & Taylor, 2015), as well as Giraffatitan (D’Emic, 2012). The undivided distal articular surface is gently convex anteroposteriorly, but does not extend onto the anterior surface of the humerus, differing from the condition in derived titanosaurs (Wilson, 2002). There is no notable beveling of the distal end.

Ulna

Both ulnae are largely complete (Figs. 20 and 21), although the left element (DAM 30) is the better preserved and least compressed of the two (see Table 5 for measurements). In both ulnae, the anterolateral proximal process has undergone crushing, resulting in a slight medial deflection in the left ulna, and prominent medial deflection in the right ulna (DAM 29).

The ratio of the maximum diameter of the proximal end to ulna length is 0.35. This is similar to most eusauropods, although the ulna is much more robust in derived titanosaurs (Wilson, 2002; Curry Rogers, 2005), as well as the basal macronarians Haestasaurus and Janenschia (Mannion et al., 2013). The olecranon process is moderately developed, but this is not a prominent process, and it does not project far beyond the remainder of the proximal articular surface, contrasting with the condition in titanosaurs (McIntosh, 1990) and some basal macronarians (Haestasaurus and Janenschia (D’Emic, 2012; Upchurch, Mannion & Taylor, 2015)). The anteromedial proximal process is longer than the anterolateral process (ratio =1.35). This value is similar to many sauropods, but is considerably lower than in the brachiosaurids Cedarosaurus and Venenosaurus (Mannion et al., 2013; Upchurch, Mannion & Taylor, 2015). In anterior view, the articular surface of the anteromedial proximal process is not concave, differing from the morphology seen in most titanosaurs (Upchurch, 1995; Upchurch, 1998), as well as some non-titanosaurian taxa, e.g., Giraffatitan (D’Emic, 2012). The posterior proximal process is much smaller than the other proximal processes, lacking the ‘T’ shaped proximal end outline that characterizes many titanosaurs (Upchurch, Mannion & Taylor, 2015).

The radial fossa along the proximal half of the ulna is striated and rugose, but there is no prominent ridge or tubercle. The proximal processes continue distally as rounded ridges, and the posterior surface of the proximal half is gently concave transversely in between. There is a prominent, proximodistally elongate interosseus crest present on the anteromedial surface: this begins a little below midheight of the ulna, and extends down to a short distance above the distal end.

The posterior surface of the distal end is flat, lacking the ridge and groove structure present in some turiasaurs (Royo-Torres, Cobos & Alcalá, 2006). The distal end is posteriorly expanded, as is the case in most sauropods, with the exception of some derived titanosaurs and additional taxa such as Giraffatitan (D’Emic, 2012). In distal view, the ulna has a D-shaped outline, with a gentle anteromedial fossa, as in most eusauropods (D’Emic, 2013; Upchurch, Mannion & Taylor, 2015).

Carpus

The complete right carpal (DAM 19; Fig. 22; Table 5) has approximately subequal mediolateral and anteroposterior dimensions (ratio =1.14), as is the case in most other basal macronarians and turiasaurs (Royo-Torres et al., 2014), although the carpal of at least one individual of Camarasaurus is much shorter anteroposteriorly (Tschopp et al., 2015). It decreases in dorsoventral height anteromedially. The proximal articular surface is concave centrally, becoming irregularly convex towards its margins. A gentle, channel-like fossa excavates the anterior surface. The distal surface is anteroposteriorly and transversely convex (although this becomes concave anteromedially), and it lacks the medial distal process that characterises Turiasaurus (Royo-Torres, Cobos & Alcalá, 2006) and Tazoudasaurus (Allain & Aquesbi , 2008).

Manus

All manual elements are described as if they were held in a horizontal position, rather than the in vivo vertical position, and metacarpals are orientated with the long axis of the distal end oriented mediolaterally. Left metacarpals I–IV, right metacarpal III, and four left manual phalanges are preserved (Figs. 23–28). Identification of the phalanges as manual, and their position within the hand, was based on comparisons with other eusauropods preserving complete manüs (e.g., Apatosaurus, Camarasaurus, Giraffatitan (Janensch, 1922; Gilmore, 1936; Tschopp et al., 2015)). Measurements are provided in Table 6. Two ungual claws were also interpreted as manual elements (Lapparent, 1943), but cannot be located in the MNHN collections.

Table 6 Measurements of the left manus of Vouivria damparisensis n. gen. n. sp. (MNHN.F.1934.6 DAM 15–18 and 20–23).

Dimension	McI	McII	McIII	McIV	I-1	II-1	III-1	IV-1	
Maximum proximodistal length	–	383	394	360	69	77	68	61	
Proximal end maximum diameter	145	105	105	98	85	96	92	71	
Proximal end diameter perpendicular  to maximum diameter	69	74	71	85	63	49	44	–	
Midshaft dorsoventral height	–	43	48	36	–	–	–	–	
Midshaft mediolateral width	–	69	54	50	66	78	76	56	
Distal end dorsoventral height	–	68	73	56	–	–	–	–	
Distal end mediolateral width	–	116	102	96	82	88	88	71	
Notes.

All measurements are in millimetres.

In articulation, the metacarpus appears to have the derived eusauropod condition (Upchurch, 1998; Wilson & Sereno, 1998; Bonnan, 2003; Allain & Aquesbi , 2008), with a semi-tubular, ‘U’-shaped outline in proximal view (Fig. 27). Metacarpal III is the longest of the complete metacarpals, followed by metacarpal II and then metacarpal IV, although there is a great deal of variation between taxa (Poropat et al., 2015a; Tschopp et al., 2015). The ratio of the length of metacarpal III to that of the ulna is 0.43. Given that the ulna is usually approximately the same length as, or slightly longer than, the radius, this indicates that the longest metacarpal to radius length ratio was greater than 0.4, consistent with a placement within Macronaria (Wilson & Sereno, 1998; Upchurch, Barrett & Dodson, 2004).

Only the proximal half of the left metacarpal I (DAM 20; Fig. 23) is preserved, and it has undergone some compression. It has a dorsoventrally tall, D-shaped proximal end outline, with a convex medial margin and concave lateral margin. It lacks the mediolateral compression seen in some titanosaurs (Mannion & Calvo, 2011). The proximal articular surface is convex.

The left metacarpal II (DAM 21; Fig. 24) is complete, but is missing small amounts of material from its shaft, and has undergone some dorsoventral compression. It is a gracile element, with a minimum shaft width to length ratio of 0.18, contrasting with the robust second metacarpals of diplodocoids and some derived titanosaurs (Sekiya, 2011; Poropat et al., 2016). The proximal articular surface is flat to very mildly convex. In proximal view, the metacarpal has a dorsoventrally tall pentagonal outline, with a flat lateral margin and convex medial margin. This medial convexity is formed by a prominent, rounded dorsomedial ridge that extends the full length of the metacarpal, becoming especially prominent approximately two-thirds from the proximal end. In general, the dorsal surface of the metacarpal is gently convex transversely, with no distinct lateral surface; instead it curves into the ventral surface. In contrast, there is a distinct medial surface for articulation with metacarpal I that faces ventromedially, and that is offset from the ventral surface by a low, rounded ridge. The mid-diaphysis has a subtriangular cross section. In distal view, the metacarpal is dorsoventrally taller along its medial half. The distal articular surface is transversely concave and dorsoventrally convex, and extends onto the dorsal surface, with a strong medial bias; however, this dorsal extension might be a result of the dorsoventral crushing of the metacarpal. The ventral margin of the distal end is prominently concave along the midline, with this extending a short distance proximally as a gentle ventral excavation. There is no notable beveling of the distal end.

Both third metacarpals are largely complete in terms of length, but the left metacarpal (DAM 22; Fig. 25) is missing material from the lateral margin of the shaft, close to the proximal end, whereas the strongly dorsoventrally compressed right element (DAM 24) has been broken into several pieces and partially glued back together, with material missing in places. The following description is therefore based on the better preserved left element, although the right element does not differ anatomically in any notable way. Although the proximal end of metacarpal III has almost certainly been affected by crushing, it has a mediolaterally wide trapezoidal shape, with a ratio of mediolateral width to dorsoventral height of 1.48. Whereas in most eusauropods this ratio is approximately 1.0, DAM 22 shares this mediolaterally expanded morphology with the brachiosaurids Giraffatitan and Venenosaurus (and, to a lesser extent, the somphospondylan Wintonotitan), although it is also present in the non-neosauropod eusauropods Atlasaurus and Jobaria (Table 7). A ridge extends along the medial margin of the ventral surface, fading out just distal to two-thirds of the metacarpal length from the proximal end. Crushing has resulted in this ridge being deflected onto the medial surface. Within the concavity formed above this ridge, there is a bulge-like process dorsal to the ventromedial ridge, at approximately one-quarter of the length from the proximal end: this is present on both the left and right metacarpal III and in both instances it appears that the ventromedial ridge bifurcates at this point, with one ridge extending along the ventromedial margin, and the other forming this bulge. We regard this as an autapomorphy of Vouivria. The medial margin of the distal end is flat, with a break of slope where it curves into the dorsal surface; the lateral margin slopes to face slightly ventrally, and is gently concave dorsoventrally. The distal articular surface is strongly concave transversely towards its centre, and gently convex dorsoventrally, with only a gentle expansion onto the dorsal surface. There is a prominent midline ventral notch at the distal end. The distal end is gently beveled, with the lateral distal condyle extending further distally than the medial condyle.

Table 7 Ratios of maximum mediolateral width to dorsoventral height of the proximal end of metacarpal III for a range of eusauropod taxa, arranged in order of increasing values.

Brachiosaurid taxa are emboldened.

Taxon and reference	Ratio	
Bellusaurus sui (Mo, 2013)	0.84	
Camarasaurus sp. (Bonnan, 2003)	0.87	
MfN MB.R.2093 (PD Mannion, pers. obs., 2012)	0.87	
Mamenchisaurus youngi (Ouyang & Ye, 2002)	0.95	
Diamantinasaurus matildae (Poropat et al., 2015a)	0.97	
Rapetosaurus krausei (Curry Rogers, 2009)	0.98	
Ligabuesaurus leanzai (MCF-PHV 233: PD Mannion, pers. obs., 2009)	0.99	
Apatosaurus louisae (Gilmore, 1936)	1.00	
Zby atlanticus (Mateus, Mannion & Upchurch, 2014)	1.05	
Angolatitan adamastor (Mateus et al., 2011)	1.08	
Omeisaurus tianfuensis (He, Li & Cai, 1988)	1.08	
Epachthosaurus sciuttoi (Martínez et al., 2004)	1.09	
Wintonotitan wattsi (Poropat et al., 2015b)	1.22	
Venenosaurus dicrocei(Tidwell, Carpenter & Meyer, 2001)	1.33	
Atlasaurus imelakei (Monbaron, Russell & Taquet, 1999)	1.43	
Giraffatitan brancai(Janensch, 1922)	1.47	
Vouivria damparisensis (this study)	1.48	
Jobaria tiguidensis (Sereno et al., 1999)	1.86	

The left metacarpal IV (DAM 23; Fig. 26) is fairly complete, but has undergone a little deformation, and the distal end is slightly damaged along its dorsal and ventral margins. In proximal view, the metacarpal forms a right-angled chevron shape, with a well-developed concavity along the ventrolateral margin for reception of metacarpal V. As such, the proximal end of metacarpal IV forms the same morphology as that observed in brachiosaurids (D’Emic, 2012), although a number of non-brachiosaurid taxa also have this chevron-like shape (Mannion et al., 2013). Throughout the metacarpal length, the dorsal surface is fairly flat. The ventromedial margin of the proximal third forms a flange, creating a concavity on the medial surface directly above; the flange is clearly a genuine feature, which we consider autapomorphic, although the concavity might be a result of deformation of this flange. Along the proximal two-thirds, the metacarpal has a semicircular cross section, with a flat ventrolateral surface that becomes increasingly ventrally-facing distally, such that by the distal third the cross section has become a transversely elongate ellipse, lacking distinct lateral or medial surfaces until the distal end. There is a weakly developed, rounded ridge on the lateral surface of the middle third of the metacarpal. The medial margin of the distal end is flat, whereas the lateral margin is dorsoventrally convex. In distal view, the metacarpal is dorsoventrally taller towards its lateral margin, and it lacks the pointed medial and lateral projections that characterize some titanosaurs (Poropat et al., 2016). There is a well-developed midline ventral concavity at the distal end, and this extends a short distance proximally along the ventral surface. The distal articular surface is transversely concave and mostly convex dorsoventrally, although it does not extend strongly onto the dorsal surface of the metacarpal. Consequently, the metacarpals show the derived, titanosauriform condition, whereby the distal articular surfaces are largely restricted to the distal end (Salgado, Coria & Calvo, 1997; D’Emic, 2012). There is no beveling of the distal end.

Phalanx I-1 (DAM 16; Figs. 28A–28C) has a transversely elongate oval shape in proximal view, with the dorsoventral height decreasing laterally. The proximal articular surface is flat to very mildly concave. In dorsal view, the lateral and (especially) medial margins of the phalanx are concave, and the medial margin is proximodistally longer than the lateral margin, meaning that the distal end is beveled. The phalanx also decreases in dorsoventral height distally. The dorsal surface is gently concave proximodistally. It is flat transversely, but becomes convex towards the medial and lateral margins, neither of which forms a distinct surface. The ventral surface is transversely convex and fairly flat proximodistally. In distal view, the medial margin is dorsoventrally taller than the lateral one. The distal articular surface is strongly convex dorsoventrally, curving onto the dorsal and (especially) ventral surfaces. Each condyle is transversely convex, with a resultant concavity in between the two condyles, along with a prominent ventral midline concavity.

Although accentuated by dorsoventral crushing, phalanx II-1 (DAM 15; Figs. 28D–28F) has a transversely elongate, semi-circular outline (with a flat ventral margin) in proximal view, and the proximal articular surface is strongly concave centrally. In dorsal view, the medial and (especially) lateral margins are concave, and there is a ventrolateral projection at the distal end. The dorsal surface is mildly concave proximodistally, and transversely convex, curving into the medial and lateral surfaces. In contrast, the ventral surface is fairly flat proximodistally, but gently concave transversely. In distal view, the medial distal condyle is larger both mediolaterally and dorsoventrally than the lateral distal condyle. The distal articular surface is strongly convex dorsoventrally, curving onto the dorsal and (especially) ventral surfaces. It is saddle-shaped transversely, being convex towards the lateral and medial margins, but strongly concave centrally, in between the two condyles. There is a prominent ventral midline concavity. The medial margin of the distal end is a close fit to the lateral margin of the distal end of phalanx I-1.

Aside from being smaller, phalanx III-1 (DAM 17; Figs. 28G–28I) does not differ in any notable way from manual phalanx II-1, including the presence of a prominent ventrolateral projection at the distal end. It is the same mediolateral width as the distal end of metacarpal III.

Phalanx IV-1 (DAM 18; Figs. 28J–28L) is much smaller than the other manual phalanges. Although broken ventrally, the proximal end is transversely elongate, with a gently convex dorsal margin. In dorsal view, the lateral and medial margins are strongly concave. The distal articular surface is transversely convex, and there is no central concavity delimiting medial and lateral condyles, although there is a midline ventral notch. There is no distinct ventrolateral projection at the distal end, although the phalanx is transversely expanded on both medial and lateral margins at the distal end. It is mediolaterally shorter than the distal end of metacarpal IV, as is the case in most other sauropods, e.g., Giraffatitan (Janensch, 1922).

Although lost, the two ungual claws were figured in medial/lateral view by Dorlodot (1934: fig. 16) and Lapparent (1943: pl. IV, figs. 13, 14), demonstrating their recurved morphology. Lapparent (1943) also commented that each one was 140 mm long. Following the interpretation of these previous authors, these elements would be the left and right first digit ungual claws. However, it remains possible that one or both elements are pedal, rather than manual, ungual claws. Comparison of the lengths of these unguals with those of the complete metacarpals and metatarsals does not help to resolve this issue. If they belong to the manus, then this demonstrates that the ungual claws were greatly reduced, with ratios of less than 0.4, as would be expected for a titanosauriform (Mannion et al., 2013). However, the lengths of the unguals are also compatible with those of the preserved metatarsals, based on comparisons with basal titanosauriforms such as Cedarosaurus (Tidwell, Carpenter & Brooks, 1999) and Tastavinsaurus (Canudo, Royo-Torres & Cuenca-Bescós, 2008). As such, we exclude the ungual claws from our phylogenetic character scoring.

Ilium

The left ilium (DAM 33) preserves the incomplete preacetabulum, pubic peduncle and acetabulum (Fig. 29; Table 8). Its internal tissue structure is not camellate, differing from the condition in derived somphospondylans (Powell, 2003; Wilson & Upchurch, 2009). The preacetabulum is incomplete along its ventral margin, and is missing most of the dorsal margin, but it clearly projects anteriorly and is also deflected laterally. This lateral deflection is less than 45°, but this might have been affected by crushing. It is not possible to determine whether there was a ventral bulge on the preacetabulum, such as that observed in Giraffatitan and several derived titanosaurs (D’Emic, 2012), because the ventral margin is too incomplete, but it can be ascertained that there was no horizontal ventral shelf, such as that which characterizes many titanosaurs (McIntosh, 1990). Based on the relatively complete ventral portion of the anterior margin of the preacetabulum, this did not taper to a point, but instead formed the semi-circular outline that characterizes Titanosauriformes (Calvo & Salgado, 1995; Upchurch, 1998; Wilson & Sereno, 1998).

Table 8 Measurements of the pelvic girdle elements of Vouivria damparisensis n. gen. n. sp. (MNHN.F.1934.6 DAM 31, 33–35).

An asterisk denotes a measurement based on an incomplete element. Note that the left ischium measurements are more reliable than those for the distorted right ischium.

Element	Dimension	Measurement	
Left ilium	Anteroposterior length as preserved	555*	
	Dorsoventral height as preserved	518*	
	Anteroposterior length of pubic peduncle at base	56	
Left pubis (?)	Proximodistal length as preserved	197*	
	Distal end anteroposterior length	182	
	Distal end maximum mediolateral width	75	
Left ischium	Iliac peduncle anteroposterior length	121	
	Iliac peduncle mediolateral width	72	
	Proximal plate anteroposterior length	183	
	Pubic articulation approximate dorsoventral height	277	
Right ischium	Proximodistal length	765	
	Iliac peduncle anteroposterior length	139	
	Iliac peduncle mediolateral width	53	
	Proximal plate anteroposterior length	188	
	Pubic articulation approximate dorsoventral height	257	
	Distal blade dorsoventral height	85	
	Distal blade mediolateral width	40	
Notes.

All measurements are in millimetres.

The ilium becomes transversely thin around the area where the pubic peduncle and preacetabulum meet. This results in the anterior margin of the upper portion of the pubic peduncle forming a sharp ridge that continues anterodorsally along the preserved ventral margin of the preacetabulum, thickening anteriorly. The pubic peduncle is complete in terms of length, but most of its medial surface has been sheared off; although it is mediolaterally wider than anteroposteriorly long, it is not possible to determine its distal end morphology. It is oriented perpendicular to the long axis of the ilium, as in other titanosauriforms (Salgado, Coria & Calvo, 1997). The medial surface of the ilium preserves a number of ridges for articulation with sacral ribs.

Pubis (?)

There is a poorly preserved distal end of a limb or pelvic bone (DAM 31) that is here interpreted as the distal end of a left pubis (Fig. 30; Table 8). The distal articular surface is anteroposteriorly convex, becoming strongly convex towards the anterior margin, without forming an anterior boot, such as that seen in Giraffatitan and Tastavinsaurus (Canudo, Royo-Torres & Cuenca-Bescós, 2008; Mannion et al., 2013). The lateral surface is not preserved along the distal two-thirds of the preserved element, so we cannot determine whether the distal pubis was transversely expanded distally. However, it seems most likely that the pubis had an unexpanded, laminar distal blade, such as that seen in titanosauriforms and rebbachisaurids (Curry Rogers, 2005; Poropat et al., 2016). As a result of our uncertainty in its identification, this element is not included in our phylogenetic character scoring.

Ischium

Both ischia are preserved (Figs. 31 and 32; Table 8). The right ischium (DAM 35) is fairly complete, aside from missing much of the anteroventral margin of the shaft. It has also undergone some mediolateral crushing. The left ischium (DAM 34) is less complete (missing the distal half of the blade and damaged around the area where the lateral ridge is present), but has not been affected by crushing.

The iliac peduncle is anteroposteriorly elongate relative to the anteroposterior length of the proximal plate (ratio =0.66), comparable to the values in several brachiosaurids (Wilson, 2002; D’Emic, 2012), as well as several other eusauropod taxa (Mannion et al., 2013). Although anteroposteriorly elongate, it is not mediolaterally compressed, and its gently convex articular surface is oval, narrowing mediolaterally towards the acetabulum. The dorsoventrally short iliac peduncle lacks the distinct “neck” that characterizes some rebbachisaurids (Sereno et al., 2007). The acetabulum merges smoothly into the lateral surface of the ischium, lacking a distinct break of slope. It also seems to retain the same mediolateral width throughout its length, contrasting with the centrally constricted morphology present in most rebbachisaurids (Mannion et al., 2012). There is a very slight anterodorsal projection of the pubic articulation, but the acetabulum lacks the strongly concave profile seen in many somphospondylans (D’Emic, 2012). The pubic articulation decreases in mediolateral thickness distally.

The lateral ridge, for attachment of M. flexor tibialis internus III, extends from the lower portion of the proximal plate and continues distally onto the upper portion of the blade. It is situated close to the posterior/dorsal margin of the ischium, and is a prominent proximodistally elongate, rounded structure. There is a gentle, parallel groove posterior to the ridge. The presence of this groove is the plesiomorphic sauropod condition, but it is lost in most titanosauriforms (D’Emic, 2012; Poropat et al., 2016).

The angle formed between the long axis of the shaft and the acetabular line (i.e., the straight line from the anterodorsal corners of the iliac and pubic peduncles) is greater than 80°, in contrast to most rebbachisaurids (Carballido et al., 2012). The upper portion of the blade is clearly twisted relative to the proximal plate, but not enough is preserved of the left ischium (and the right ischium is too distorted) to be able to determine the nature of the conjoined ischia morphology. The distal end of the blade does not become triangular, but instead remains blade-like, and the blade is gracile throughout, with a maximum blade width to ischium length ratio of 0.11. This is comparable to the low values seen in many basal macronarians (Upchurch, 1998; Mannion et al., 2013).

Femur

Aside from damage to the fibular distal condyle, the right femur (DAM 36) is fairly complete (Fig. 33; see Table 9 for measurements). However, it is currently in two pieces, anteroposteriorly crushed, and the anterior surface is heavily deformed and worn in places. An incomplete and poorly preserved portion of the distal end of a left femur is preserved, with heavily worn condyles. A portion of large limb bone might represent a more proximal portion of this left femur.

Table 9 Measurements of the upper hindlimb elements (femur, tibia, fibulae and astragalus) of Vouivria damparisensis n. gen. n. sp. (MNHN.F.1934.6 DAM 11, 36–40).

For the fibulae measurements, the first value represents the left element (DAM 40) and the second value the right element (DAM 39). An asterisk denotes a measurement based on an incomplete element.

Element	Dimension	Measurement	
Right femur	Proximodistal length	1,460	
	Distance from proximal end to distal tip of fourth trochanter	670	
	Midshaft mediolateral width	239	
	Midshaft anteroposterior length	83	
	Midshaft minimum circumference	600	
	Distal end maximum anteroposterior length as preserved	160*	
	Distal end mediolateral width (tibial condyle)	168	
	Distal end mediolateral width (fibular condyle)	210	
Left tibia	Proximodistal length	860	
	Proximal end maximum anteroposterior length	160	
	Proximal end maximum mediolateral width (excluding cnemial crest)	291	
	Transverse width of cnemial crest (measured along posterior surface)	77	
	Midshaft maximum diameter	139	
	Midshaft diameter perpendicular to maximum diameter	65	
	Distal end mediolateral width	246	
	Distal end maximum anteroposterior length	116	
Fibulae	Proximodistal length	870*/890	
	Proximal end anteroposterior length	213/221	
	Proximal end maximum mediolateral width	79/53	
	Distance from proximal end to distal tip of lateral trochanter	430/428	
	Midshaft anteroposterior length	93/95	
	Midshaft mediolateral width	55/50	
	Distal end anteroposterior length	–/167	
	Distal end mediolateral width	–/119	
Astragalus	Maximum proximodistal height	102	
	Maximum mediolateral width	234	
	Maximum anteroposterior length	145	
Notes.

All measurements are in millimetres.

The femoral head projects dorsomedially, similar to a number of other macronarians (Curry Rogers, 2005; Poropat et al., 2016), although contrasting with the medially-directed femoral head of Brachiosaurus and Giraffatitan (Riggs, 1903a; Janensch, 1961). As in most macronarians, the lateral margin of the proximal end, above the lateral bulge, is medially deflected relative to the lateral margin of the midshaft (Mannion et al., 2013). There is no lateral trochanteric shelf or associated ridge on the posterior surface, contrasting with many titanosaurian (Otero, 2010; Mannion et al., 2013) and rebbachisaurid (Sereno et al., 2007) femora. It is not possible to determine whether the linea intermuscularis cranialis ridge, a feature of several derived titanosaurs (Otero, 2010; D’Emic, 2012; Poropat et al., 2015b), is definitely absent on the anterior surface of the shaft.

The fourth trochanter is a well-developed structure, positioned on the medial margin of the posterior surface. It does not extend to femoral midlength, contrasting with most sauropods, but is similar to the condition seen in Giraffatitan (Upchurch, 1998). The fourth trochanter has been crushed, such that its posterior surface faces medially; however, despite this crushing, it is still not visible in anterior view. As such, it differs from the anteriorly visible fourth trochanters of other brachiosaurids (Mannion et al., 2013) and several additional taxa (Whitlock, 2011a). The medial surface, directly anterior to the fourth trochanter, forms a channel-like groove, although this has been accentuated by crushing.

The mediolateral width of the tibial to fibular condyle is 0.8. This low value is comparable to that of Brachiosaurus and Giraffatitan, as well as many somphospondylans (Poropat et al., 2016), whereas most other sauropods have ratios closer to 1.0 (Wilson, 2002). The fibular distal condyle is divided into two distinct condyles, as in most derived eusauropods (Sekiya, 2011). It is not possible to determine whether the distal condyles extended strongly onto the anterior surface of the femur. Two narrow, intercondylar ridges are present in between the tibial and fibular condyles, at the distal margin of the posterior surface (Fig. 33). We have not observed these ridges in other sauropod femora, and thus tentatively regard them as an autapomorphy of Vouivria. There is a slight beveling of the distal end of the femur, with the tibial distal condyle extending further distally than the fibular condyle. Such beveling is present amongst an array of sauropods (e.g., Shunosaurus, Diplodocus, Cedarosaurus, Tastavinsaurus; Mannion et al., 2013), but differs from the non-beveled morphology of Brachiosaurus and Giraffatitan (Riggs, 1903a; Janensch, 1961).

Tibia

The left tibia is complete (DAM 37; Fig. 34; Table 9), but has undergone some anteroposterior compression, including a crushed section on the anterior surface (just above midheight) that has formed a fairly prominent concavity. A poorly preserved fragment appears to be the distal end of the right tibia (DAM 38). The tibia to femur length ratio is 0.59. In most derived eusauropod taxa this value is 0.6 or greater, but several somphospondylans have ratios that are less than 0.6 (D’Emic et al., 2013).

As preserved, the proximal and distal ends are transversely elongate and anteroposteriorly compressed, but this has undoubtedly been accentuated by crushing. Although it might have been more anteriorly directed prior to crushing, the laterally directed cnemial crest clearly did not project entirely anteriorly. The lateral margin of the cnemial crest is slightly incomplete, meaning that we cannot determine whether it formed a triangular point or remained rounded. There is a ‘tuberculum fibularis’ (see Harris, 2007) on the posterior (internal) surface of the cnemial crest. A similar feature is present on the tibia of Janenschia (SMNS 12144: PD Mannion, pers. obs., 2011) and Giraffatitan (MfN MB.R.2181.46 [SII]: PD Mannion, pers. obs., 2014), as well as several flagellicaudatan taxa (Harris, 2007; Tschopp, Mateus & Benson, 2015). A ‘second cnemial crest’ is also present as a small pointed projection. This feature is present in many eusauropod tibiae, but is lost in most somphospondylans and diplodocoids (Mannion et al., 2013). The posterior margin of the proximal end is gently concave.

The anterior surface of the tibia is mainly gently convex transversely, becoming flat at the distal end. Both the anteromedial and anterolateral margins of the distal end form sharp ridges, but these are likely to have been accentuated, or even fully caused, by crushing. Although the medial distal condyle extends further distally than the lateral condyle, it is reduced and does not extend as far laterally. The mediolateral width of the distal end is less than twice that of the maximum diameter of the midshaft (ratio =1.77), contrasting with the expanded distal ends of many titanosaurian tibiae (Wilson, 2002).

Fibula

Both fibulae are preserved (Fig. 35; see Table 9 for measurements). The right fibula (DAM 39) is complete, aside from a piece missing from the posterior margin near the proximal end, but has been transversely compressed. The uncrushed left fibula (DAM 40) is complete except for the distal end, which is missing material from the medial and lateral surfaces, and is damaged along the anterior margin; it might also be missing a small amount of material from the distal articular surface, which would also explain why it is 20 mm shorter than the right fibula.

In lateral view, the fibula is straight, rather than having the sinuous morphology that characterizes many somphospondylans (Canudo, Royo-Torres & Cuenca-Bescós, 2008; D’Emic, 2012). The medial surface is flat and the lateral surface is anteroposteriorly convex for most of the fibula length. There is no anteromedial crest at the proximal end, contrasting with most somphospondylans (Wilson & Upchurch, 2009; D’Emic, 2012; Mannion et al., 2013), nor is there an anterolateral trochanter, such as that observed in several titanosaurs (González Riga, 2003; Wilson, Barrett & Carrano, 2011). As in other neosauropods (Wilson & Sereno, 1998), the medial surface of the proximal end forms a raised, striated region.

The lateral trochanter is a low and singular structure, with a gentle groove situated anteriorly. There is a slightly raised area anterior to this groove on the right fibula, but this is absent in the better preserved left fibula, suggesting that this morphology has been affected by crushing. As such, in contrast to many somphospondylans (Upchurch, 1998; Mannion et al., 2013), the lateral trochanter is not composed of parallel ridges. Unlike most non-diplodocid eusauropods (Whitlock, 2011a), the lateral trochanter does not extend distally to the fibular midheight.

The anterior margin of the distal third forms a sharp ridge, and the elliptical-shaped distal end is expanded transversely (as well as slightly anteroposteriorly), with a well-developed medial, lip-like expansion. Distally, the fibula is twice as thick mediolaterally as the midshaft, as is the case in Giraffatitan (Wilson, 2002) and a small number of somphospondylans (Mannion et al., 2013), including Tastavinsaurus (Canudo, Royo-Torres & Cuenca-Bescós, 2008).

Tarsus

The left astragalus (DAM 11; Fig. 36) is complete and well preserved (Table 9). In articulation, the astragalus caps most of the distal end of the tibia, contrasting with the reduced astragalus that characterizes other titanosauriforms (Wilson & Upchurch, 2009; Ksepka & Norell, 2010). It is wedge-shaped in dorsal view, decreasing in anteroposterior length medially along the posterior margin, as is the case in all derived eusauropods (Upchurch, 1995; Upchurch, 1998).

The astragalus of Vouivria also decreases in dorsoventral height medially. The ventral surface is strongly convex anteroposteriorly, and gently convex transversely. Although the lateral surface is dorsoventrally concave, this does not form a ventrolateral lip. Such a lip characterizes the astragali of most sauropods, but is absent in the majority of macronarians (Wilson & Upchurch, 2009; Mannion et al., 2013). Unlike the condition in diplodocoids (Whitlock, 2011a), the lateral surface is not deflected posteriorly. Shallow fossae excavate the lateral surface within the central concavity.

The ascending process extends to the posterior margin of the astragalus, a condition that characterizes Neosauropoda (Wilson & Sereno, 1998; Wilson, 2002). Mannion & Otero (2012) provided evidence that the titanosaur Elaltitan had autapomorphically reversed this condition; however, re-examination of this left (not right) element (PVL 4628: PD Mannion pers. obs., 2013) reveals that it was misinterpreted, and the deformed ascending process does in fact extend to the posterior margin. The posterior surface of the ascending process is strongly concave dorsoventrally, although the depth of this concavity might have been accentuated by crushing. There is a small posterior process posteromedial to the ascending process, as is the case in many sauropods, but which is lost in most somphospondylans (D’Emic, 2012; Mannion et al., 2013). A ridge extends posteromedially from the posteromedial corner of the ascending process, and there is a large fossa anteromedial to this ridge. Medial to the ascending process, the dorsal surface of the astragalus is fairly flat and featureless, and tilted to face slightly posteriorly, as well as dorsally.

No calcaneum is preserved, but we cannot determine whether this is a genuine absence, or merely a taphonomic artefact. A calcaneum is present in most sauropods for which a well preserved lower hindlimb and metatarsus is known, but is lost in titanosaurs (McIntosh, 1990; Upchurch, 1995; Upchurch, 1998; Bonnan, 2000). A possible calcaneum was described in the titanosaur Elaltitan (Mannion & Otero, 2012), but re-examination of this element (PVL 4628: PD Mannion, pers. obs., 2013) reveals that it is actually a portion of the distal end of the tibia, adhered to the fibula.

Pes

A left (DAM 12; Fig. 37) and right metatarsal I (DAM 13) are preserved, although the right element is incomplete along its medial and lateral margins, and both elements are compressed. The dorsal surface of the distal end of the left element is abraded. Only the distal end of the left metatarsal II is preserved, and the complete left metatarsal III (DAM 14; Fig. 38) is strongly compressed dorsoventrally, with the proximal end especially affected. Measurements are provided in Table 10.

Table 10 Measurements of the left metatarsals of Vouivria damparisensis n. gen. n. sp. (MNHN.F.1934.6 DAM 12 and 14).

An asterisk denotes a measurement based on an incomplete element.

Dimension	MtI	MtIII	
Maximum proximodistal length	175	234	
Proximal end maximum diameter	109	123	
Proximal end diameter perpendicular to maximum diameter	95	42	
Midshaft dorsoventral height	39	28	
Midshaft mediolateral width	77	64	
Distal end maximum dorsoventral height	48*	53	
Distal end maximum mediolateral width	120	75	
Notes.

All measurements are in millimetres.

The proximal end of metatarsal I has a D-shaped outline, with a concave lateral margin and a dorsolateral, tapered projection. This cross section is maintained along the proximal half, before there is a strong twist in the metatarsal axis distally. As in most eusauropods (Wilson, 2002), the articular surface of the proximal end is beveled to face medially relative to the long axis of the shaft. There are no foramina present on the dorsal surface, and no dorsolateral rugosity close to the distal end, such as that found in diplodocids (Upchurch, 1995). The distal end is ventrolaterally expanded, such that it extends further laterally than the proximal end, although this might be affected by crushing. Regardless, this feature appears to be phylogenetically ‘plastic’ within Eusauropoda (Upchurch, 1998; D’Emic, Wilson & Williamson, 2011; Mannion et al., 2013). The distal end is beveled, as is the case in most eusauropods (Wilson, 2002), with the dorsoventrally taller lateral distal condyle extending further distally than the medial one. There is a prominent ventral notch along the distal end midline, and the distal articular surface of each condyle is transversely convex, resulting in a central concavity.

The distal end of metatarsal II is dorsoventrally tallest on its medial half. The lateral surface of the distal end slopes to face dorsolaterally, with the medial surface of the distal end of metatarsal III presumably resting on top of this surface. In contrast, the medial surface of the distal end is concave, with the dorsomedial margin forming a sharp ridge; the ventrolateral expansion of the distal end of metatarsal I fits into this concavity. The distal articular surface extends onto the dorsal surface of the metatarsal, but does not extend far proximally.

The metatarsal III to tibia length ratio is 0.27. In most non-titanosauriforms this value is less than 0.25 (Wilson & Sereno, 1998). In dorsal view, the medial margin of metatarsal III is strongly concave, with a much straighter lateral margin. It is not possible to determine whether the proximal articular surface is domed, as is the case in several macronarian taxa (D’Emic, Wilson & Williamson, 2011). There is a short, low, rounded ridge on the lateral margin, at approximately midlength. There is no dorsolateral rugosity at the distal end. The medial distal condyle extends very slightly further distally than the lateral condyle, but this might have been caused by crushing. The distal articular surface is dorsoventrally convex and transversely concave, and extends onto the dorsal surface of the metatarsal. There is a ventral midline notch at the distal end. The dorsoventrally taller lateral margin of the distal end is gently concave as a result of a small ventrolateral projection, whereas the shorter medial margin curves into the dorsal margin.

Phylogenetic Analysis

Dataset

We have revised the scores for Vouivria damparisensis in the ‘LSDM’ data matrix of Mannion et al. (2013), using the latest augmented version of this matrix presented in Poropat et al. (2016). Emended scorings for Europasaurus follow new information presented in Carballido & Sander (2014) and Marpmann et al. (2015), and those for Lusotitan and Sonorasaurus follow Mocho, Royo-Torres & Ortega (2016) and D’Emic, Foreman & Jud (2016), respectively. We have revised scores for the Middle Jurassic Moroccan taxon Atlasaurus (Monbaron, Russell & Taquet, 1999) based on casts stored at the MNHN (cranial remains, as well as elements of the manus and pes). We have also incorporated the recently described sauropod Padillasaurus, from the late Early Cretaceous of Colombia, which was recovered as a brachiosaurid by Carballido et al. (2015); our scores for that taxon are based on the original publication, as well as additional information and photographs supplied by JL Carballido. The sauropod Jobaria is also added as an OTU, because it might be approximately contemporaneous with Atlasaurus (Middle–Late Jurassic of Niger; Rauhut & López-Arbarello, 2009), has been recovered as a close relative to that taxon in some analyses (e.g., Upchurch, Barrett & Dodson, 2004), is known from the majority of the skeleton (Sereno et al., 1999), and has been studied first-hand (MNN specimens: PD Mannion, pers. obs., 2012). We have also added in representatives of Turiasauria (Turiasaurus, Losillasaurus and Zby), given that this clade has been recovered as close to the neosauropod radiation in all previous analyses (e.g., Royo-Torres, Cobos & Alcalá, 2006; Royo-Torres & Upchurch, 2012; Carballido et al., 2015), and that its members were present in approximately contemporaneous deposits to Vouivria (i.e., the Late Jurassic–earliest Cretaceous of Europe). Our scores for Turiasauria are based on the following sources: Turiasaurus (Royo-Torres, Cobos & Alcalá, 2006; Royo-Torres & Upchurch, 2012; CPT-1195-1210: PD Mannion, pers. obs., 2009); Losillasaurus (Casanovas, Santafe & Sanz, 2001; MCNV Lo: PD Mannion, pers. obs., 2009); and Zby (Mateus, Mannion & Upchurch, 2014; ML 368: PD Mannion, pers. obs., 2012).

Nineteen additional characters have been added to the data matrix, nine of which are novel to our study. The resultant data matrix comprises 77 taxa scored for 416 characters. Characters 1–397 are the same as those presented in previous iterations of the data matrix (Mannion et al., 2013; Poropat et al., 2015a; Poropat et al., 2015b; Poropat et al., 2016; Upchurch, Mannion & Taylor, 2015), although one of these (character 40) is revised (see below). Characters 398–416 are newly added to this data matrix and presented below, including their sources (along with the character number from the original study, in square brackets). The full character list, scorings for C1–397 that differ from those presented in Poropat et al. (2016), and the data matrix (Mesquite and TNT file) are provided in Supplemental Information 1.

Revised character

C40. Humerus to femur proximodistal length ratio: 0.8 or less (0); >0.8 to <0.9 (1); 0.9–0.95 (2); >0.95 (3) (Wilson, 2002; Upchurch, Barrett & Dodson, 2004; Poropat et al., 2016; an additional state has been added here to capture variation within Titanosauriformes (ordered character)).

Added characters

C398. Basioccipital, fossa on lateral surface, extending from base of occipital condyle to base of basal tubera: absent (0); present (1) (Tschopp, Mateus & Benson, 2015: fig. 22 [C80]).

C399. Basicranium, cranial nerve opening II (optic foramen): single opening (0); medially divided to form two foramina (1) (Sander et al., 2006) [C129]; see Marpmann et al. (2015: p. 248–249 and fig. 13) for a discussion and illustration of this feature).

C400. Surangular, anterior foramen: absent (0); present (1) (Tschopp, Mateus & Benson, 2015: fig. 32A [C109]; modified here to restrict to the anterior opening).

C401. Splenial, position of anterior end relative to mandibular symphysis: posterior to symphysis (0); participates in symphysis (1) (Upchurch, 1998 [C61]).

C402. Teeth, D-shaped crown morphology in labial/lingual view: narrows mesiodistally along its apical third (0); narrows mesiodistally along its apical half, giving it a ‘heart’-shaped outline (1) (new character: based on Royo-Torres, Cobos & Alcalá, 2006; Mateus, Mannion & Upchurch, 2014: fig. 4).

C403. Middle–posterior cervical neural arches, vertical midline lamina (part of the interprezygapophyseal lamina [TPRL]) divides the centroprezygapophyseal fossa (CPRF) into two fossae: absent (0); present (1) (new character: based on Upchurch & Martin, 2002: fig. 3; Curry Rogers, 2009; PD Mannion & P Upchurch, pers. obs., 2009–2013).

C404. Middle–posterior cervical neural arches, vertical midline lamina (part of the interpostzygapophyseal lamina [TPOL]) divides the centropostzygapophyseal fossa (CPOF) into two fossae: absent (0); present (1) (new character: based on Upchurch & Martin, 2002: fig. 3; Curry Rogers, 2009; PD Mannion & P Upchurch, pers. obs., 2009–2013).

C405. Middle cervical neural spines, lateral fossa at the base of the prezygapophyseal process bounded by SPRL, PRDL and PODL: absent (0); present (1) (Harris, 2006 [C124]; Tschopp & Mateus, 2013 [C118]; see Tschopp, Mateus & Benson, 2015: fig. 51).

C406. Middle and posterior cervical neural spines, lateral surface between PRDL, PODL, SPOL (i.e., the spinodiapophyseal fossa [SDF]), has 3 or more coels separated from each other by low ridges: absent (0) present (1) (new character: based on PD Mannion & P Upchurch, pers. obs., 2009–2013; see Fig. 10A).

C407. Cervical ribs, anterior projection extends beyond anterior margin of centrum (including condyle): present (0); absent (1) (new character).

C408. Sacral ribs, Sv2 ribs: emanate solely from Sv2 (0); emanate from Sv2, with a contribution from Sv1 (1) (new character; see Fig. 14).

C409. Anteriormost caudal centra, ACDL: absent, or represented by no more than a faint ridge (0); present, well defined or sheet-like (1) (Wilson, 2002 [C129]; modified here based on PD Mannion & P Upchurch, pers. obs., 2009–2013; see Fig. 15A).

C410. Anterior–middle caudal neural arches: spinopostzygapophyseal lamina (SPOL) shape: SPOL grades smoothly toward postzygapophyses (0); SPOL abruptly ends near the anterior margin of the postzygapophyseal facet, and postzygapophyses sharply set off from neural spine, often projecting as distinct processes (1) (D’Emic, Foreman & Jud, 2016 [C121]; note that this feature is usually present in the last few anterior and first few middle caudal vertebrae; see Mannion et al., 2013: fig. 6).

C411. Anterior–middle caudal neural arches, anteroposteriorly oriented ridge and fossa (‘shoulder’) between prezygapophyses and postzygapophyses: absent (0); present (1) (D’Emic, Foreman & Jud, 2016 [C122]; note that this feature is usually present in the last few anterior and first few middle caudal vertebrae; see Mocho, Royo-Torres & Ortega, 2016: fig. 9L).

C412. Radius, proximal to distal end anteroposterior length ratio: 0.5 or greater (0); less than 0.5 (1) (new character: based on Mateus, Mannion & Upchurch, 2014: fig. 9).

C413. Ulna, vertical groove and ridge structure on posterolateral surface of distal shaft: absent (0); present (1) (Royo-Torres, Cobos & Alcalá, 2006).

C414. Carpal bones, distal carpal mediolateral width to anteroposterior length ratio: less than 1.4 (0); 1.4 or greater (1) (new character: based on Royo-Torres et al., 2014: fig. 10; note that this is the largest carpal element in those taxa with more than one carpal; see also Fig. 22).

C415. Metacarpal III, maximum mediolateral width to dorsoventral height of the proximal end ratio: less than 1.3 (0); 1.3 or greater (1) (new character; see Fig. 25E).

C416. Tibia, tubercle (‘tuberculum fibularis’) on posterior (internal) face of cnemial crest: absent (0); present (1) (Harris, 2007; Tschopp, Mateus & Benson, 2015: fig. 103 [C445]; see also Fig. 34C).

Analytical approach and results

Following previous analyses of iterations of this data matrix, characters 11, 14, 15, 27, 40, 51, 104, 122, 147, 148, 177, 195, 205 and 259 were treated as ordered multistate characters, and nine unstable and highly incomplete taxa (Astrophocaudia, Australodocus, Brontomerus, Fukuititan, Fusuisaurus, Liubangosaurus, Malarguesaurus, Mongolosaurus, Tendaguria) were excluded a priori (see Mannion et al., 2013; Poropat et al., 2016), although Sonorasaurus was retained as a putative brachiosaurid. In our parsimony analysis, we used equal weighting, rather than implied weighting (Goloboff, Mattoni & Quinteros, 2006), as the latter has been demonstrated to propagate errors and result in poorer topological accuracy in simulation studies (Congreve & Lamsdell, 2016; O’Reilly et al., 2016).

The pruned data matrix was then analysed using the ‘Stabilize Consensus’ option in the ‘New Technology Search’ in TNT vs. 1.1 (Goloboff, Farris & Nixon, 2008). Searches were carried out using sectorial searches, drift, and tree fusing, with the consensus stabilized five times, prior to using the resultant trees as the starting trees for a ‘Traditional Search’ using Tree Bisection-Reconstruction. This resulted in 72 MPTs of 1656 steps and produced a fairly well resolved strict consensus tree (Fig. 39), with no topological difference to that presented in Poropat et al. (2016), except at the base of the tree (loss of resolution between Omeisaurus, Mamenchisauridae and more derived eusauropods) and within Brachiosauridae (as well as the addition of Jobaria, Turiasauria and Padillasaurus). Jobaria and Turiasauria (Zby + (Losillasaurus + Turiasaurus)) are recovered in a polytomy with Lapparentosaurus as non-neosauropods, more basal than Atlasaurus with respect to Neosauropoda. Our topology also provides further support for the macronarian affinities of Galveosaurus (see also Barco, Canudo & Cuenca-Bescós, 2006; Carballido et al., 2011a; D’Emic, 2012; Mannion et al., 2013), differing from a position within Turiasauria, as proposed by some authors (Royo-Torres, Cobos & Alcalá, 2006; Royo-Torres & Upchurch, 2012; Mocho, Royo-Torres & Ortega, 2014).

Figure 38 Left metatarsal III of Vouivria damparisensis (MNHN.F.1934.6 DAM 14).

(A) Ventral view; (B) dorsal view; (C) lateral view; (D) medial view; (E) proximal view; (F) distal view. Scale bar equals 5 cm.

In contrast to Carballido et al. (2015), Padillasaurus is recovered as a non-titanosaurian somphospondylan. Brachiosauridae comprises the same taxa as in previous versions of this data matrix, but now consists of a paraphyletic array of Late Jurassic taxa (with Europasaurus, Vouivria and Brachiosaurus as successively more nested taxa) that lie outside of a polytomous clade comprising Lusotitan, Giraffatitan and Cretaceous brachiosaurids (i.e., Abydosaurus, Cedarosaurus, Sonorasaurus and Venenosaurus). The Pruned Trees option in TNT recognises Abydosaurus as one of the most unstable taxa; its exclusion resolves Giraffatitan and Sonorasaurus as sister taxa, forming a clade with Lusotitan + (Cedarosaurus + Venenosaurus) (Fig. 40). Bremer support values are generally low, although Brachiosauridae and the clade of brachiosaurids more derived than Europasaurus each have Bremer supports value of 2, and both Titanosauriformes and Galveosaurus + Titanosauriformes have Bremer support values of 3.

Discussion

Character correlation

Character correlation via the non-independence of characters is a potentially problematic issue for any phylogenetic analysis. If a set of characters represents non-independent aspects of a single anatomical modification, then this feature will be over-weighted in the analysis, leading to a potentially biased topology. Wilkinson (1995) identified two types of non-independent characters: ‘logically’ and ‘biologically’ non-independent. Two (or more) characters are not logically independent if the state of the first character (e.g., presence of an opening) places a constraint on that of the second character (e.g., division of this opening). However, if there is variation in the state scores for the second character (i.e., there are taxa with this opening undivided and some with it divided), then the two characters capture different information, and non-independence issues can be avoided through the use of reductive or composite coding (Wilkinson, 1995; Strong & Lipscomb, 1999), as applied here. Two (or more) characters are not biologically independent if a transformation in one is coupled with a transformation in the other by a biological process (Wilkinson, 1995), e.g., a genetic, developmental or functional interaction (Upchurch, 1999). However, recognizing such a link is difficult enough in living taxa, let alone fossil taxa, and a correlation does not necessarily mean that the characters are biologically related (Maddison & FitzJohn, 2015).

Figure 39 Strict consensus cladogram of 72 MPTs.

Brachiosauridae is highlighted in red, and non-brachiosaurid taxa previously included in the clade are highlighted in blue. Note that this tree was produced following the a priori exclusion of nine unstable taxa (see text for details).

Figure 40 Time-calibrated phylogenetic tree, showing geographic distribution of basal Macronaria.

Brachiosauridae is resolved after the a posteriori deletion of Abydosaurus in TNT. Silhouette of Giraffatitan drawn by Matthew Wedel and available at Phylopic under a Creative Commons Attribution 3.0 Unported license (https://creativecommons.org/licenses/by/3.0/legalcode). Global palaeogeographic reconstructions from Fossilworks (http://fossilworks.org/) showing the distribution of Late Jurassic and Early Cretaceous brachiosaurids.

D’Emic, Foreman & Jud (2016: p. 127) contended that many of the characters in the phylogenetic data matrix of Mannion et al. (2013) used herein are “interrelated”. Although we do not dispute that several characters might be “biologically related”, they are not consistent across taxa. For example, D’Emic, Foreman & Jud (2016) highlighted four characters pertaining to aspects of forelimb gracility (characters 42, 45, 50 and 53), arguing that “sauropods with gracile forelimbs tend to have each long bone elongated, not just one segment”. However, a survey of the scores in Mannion et al. (2013) shows just how much these scores vary between taxa, including those that are considered as closely related to one another. For example, Apatosaurus is scored as 0110 for these four characters, whereas Diplodocus is scored as 0001. Haestasaurus also has a robust humerus and ulna, but a gracile radius (Upchurch, Mannion & Taylor, 2015). Nevertheless, we re-ran our analysis with C45, C50 and C53 excluded, resulting in just one topological change (slightly better resolution in derived titanosaurs), a smaller number of MPTs (48), and fewer steps (1,637), suggesting that even if these characters are over-weighted, they have no notable effect on our topology, and their exclusion means that some of the morphological variation present between derived titanosaurs is missed. Although D’Emic, Foreman & Jud (2016) made the valid point that “subtle differences could be due to the vicissitudes of preservation or the delineation of states by a given researcher”, this cannot explain those differences that are far from subtle.

D’Emic, Foreman & Jud (2016) also commented upon the interrelation of a character capturing the depth of the supracondylar fossa on the posterior face of the distal humerus (C228) and one pertaining to the prominence of the olecranon process of the ulna (C233). Yet a small number of taxa with a deep supracondylar fossa lack a prominent olecranon process (most notably Giraffatitan (D’Emic, 2012)), and several taxa with the latter process have a shallow fossa (e.g., Haestasaurus and Janenschia (Upchurch, Mannion & Taylor, 2015); Chuanjiesaurus (Sekiya, 2011); also note that the derived titanosaur Opisthocoelicaudia has a prominent olecranon process (Borsuk-Białynicka, 1977), and yet its supracondylar fossa is of a similar depth as that of Omeisaurus (Upchurch, Mannion & Taylor, 2015)). Furthermore, the presence of a prominent olecranon process is the plesiomorphic sauropodomorph condition and yet these basal forms tend to only have a shallow supracondylar fossa (e.g., Saturnalia (Langer, França & Gabriel, 2007) and Antetonitrus (McPhee et al., 2014)). As such, it is not fully evident that the two are even biologically related.

Two other potentially interrelated characters highlighted by D’Emic, Foreman & Jud (2016) pertain to the dimensions of the articular surfaces of the pubic peduncle of the ilium (C57) and the iliac peduncle of the pubis (C58). As these two processes articulate, then their morphology might be expected to covary. However, an examination of the scores in Mannion et al. (2013) reveals the full range of combinations, with taxa scored as 00 (e.g., Camarasaurus), 01 (e.g., Opisthocoelicaudia), 10 (e.g., Tastavinsaurus) and 11 (e.g., Giraffatitan), with the quantitative ratios for several of those scores far removed from the chosen state boundary. We accept, however, that there might be clearer grounds for considering these two characters as not fully independent of one another.

D’Emic, Foreman & Jud (2016) also noted that Mannion et al. (2013) included eleven characters (C15, 16, 17, 21, 22, 25, 26, 28, 29, 30 and 31) to capture the proportions of vertebral centra, commenting that although there is “variation along the column, it does not vary independently as eleven characters”. C15 pertains to the elongation of cervical vertebrae, which is not otherwise captured in any other character, and is also included in the data matrix of D’Emic (2012). C16 and C17 pertain to the ratios of the dorsoventral to transverse diameters of the posterior articular cotyle of anterior and middle–posterior cervical centra, respectively. The plesiomorphic state for C16 (mediolaterally compressed centra) is restricted to five East Asian taxa (Shunosaurus, Omeisaurus, Mamenchisaurus, Euhelopus and Erketu), as well as Jobaria, and historically was regarded as a synapomorphy uniting many of these taxa within Euhelopodidae (e.g., Upchurch, 1998). However, the middle–posterior cervical centra of Shunosaurus are dorsoventrally compressed, and other taxa have mediolaterally compressed middle–posterior cervical centra (e.g., Nigersaurus, as well as several other diplodocoid taxa (Tschopp, Mateus & Benson, 2015)). C21 and C22 pertain to the same dimensions of the anterior and middle–posterior dorsal vertebrae, although the ratio is expressed the other way round (i.e., transverse width divided by dorsoventral height). With the exception of Omeisaurus, there does appear to be strong covariation between C17 and C21. In contrast, several scores vary between C21 and C22, with some notable differences in ratio values for a number of taxa (e.g., Apatosaurus and Giraffatitan). The remaining vertebral characters pertain to the tail. C25, C28 and C30 capture the mediolateral width to dorsoventral height ratios of the anterior, middle and posterior caudal centra, respectively. Scores for these three characters include: 000 (e.g., Omeisaurus), 010 (e.g., Tastavinsaurus), 001 (e.g., Jobaria), 100 (e.g., Camarasaurus), 110 (e.g., Apatosaurus), and 111 (e.g., Saltasaurus). C26, C29 and C31 capture the elongation of the anterior, middle and posterior caudal centra, respectively. Although each of these uses a different state boundary, there is again a great deal of variation, with 000 (e.g., Omeisaurus), 001 (e.g., Apatosaurus), 011 (e.g., Malawisaurus), 100 (e.g., Giraffatitan), 101 (e.g., Alamosaurus), and 111 (e.g., Saltasaurus). As such, although we acknowledge that the dimensions of centra can be affected by deformation (e.g., see Tschopp, Russo & Dzemski, 2013), nine of these eleven vertebral characters show evidence for a substantial degree of variation, whilst C16 and C21 show evidence for a high degree of covariation with C17. We re-ran our analysis with C16 and C21 excluded: our topology is identical, producing the same number of MPTs (72) with fewer steps (1649). We also re-ran our analysis with C16, C21, and C28–C31 excluded, which resulted in fewer MPTs (48) and steps (1623), and a gain in resolution between Jobaria, Lapparentosaurus and Turiasauria. These sensitivity analyses indicate that these vertebral characters do not bias the analysis by over-weighting certain anatomical features and, if anything, indicate that their exclusion means that some of the morphological variation present between basal eusauropod taxa is missed. Lastly, given that sauropods have >75 vertebrae (and often many more), eleven characters might be needed to capture variation in centrum proportions along the column.

Numerous authors have stressed the importance of serial variation in sauropod vertebrae (e.g., Wilson, Barrett & Carrano, 2011; D’Emic, 2012; Wilson, 2012; Wedel & Taylor, 2013; Tschopp, Mateus & Benson, 2015), and thus we contend that this information should also be captured in phylogenetic analyses. One problem with working on fossil taxa that are almost always incompletely known is that many characters cannot be scored because of missing data. For example, 33 taxa in the current version of the Mannion et al. (2013) data matrix can be scored for one or both of C16 (anterior cervical vertebrae) and C17 (middle–posterior cervical vertebrae), but only 20 of these 33 taxa can be scored for both characters. Given that we know that at least some taxa do not have the same morphology for these two characters, we caution whether it is advisable to lump them together as one character (leading to some polymorphic scores) and assume that the remaining 13 taxa have a consistent morphology throughout their cervical series.

Excluding characters (that do not fully covary) because they are ‘biologically related’ is also problematic if these skeletal changes do not all occur at one node of the tree. For example, the ‘wide-gauge’ stance of titanosaurs seems to be the product of numerous anatomical changes, including the medial deflection of the proximal femur, increased eccentricity of the femoral shaft, and bevelling of the distal femur (Wilson & Carrano, 1999). These changes might also be linked to the shortening of the tail (e.g., Wilson, 2002), reduction of caudofemoral musculature (Ibiricu, Lamanna & Lacovara, 2014), increased flaring of the preacetabular processes of the ilia (Wilson & Carrano, 1999), the enlargement and/or novel development of muscle attachments on the pectoral girdle and forelimb (Otero, 2010), and even to neck elongation via the anterior shift in centre-of-mass (Bates et al., 2016). However, these changes do not happen at a single node on the tree (e.g., see Wilson & Carrano, 1999: fig. 6), with some taxa possessing only a single derived state amongst these characters, for example, and yet they are potentially biologically related. If we excluded such characters, then we would be unable to capture the gradual assembly of the titanosaurian body plan (see also Upchurch (1999) for an example regarding cranial characters).

In summary, we argue that where there is variation, this should be captured, rather than ignored. We agree with the concern of D’Emic, Foreman & Jud (2016) that the splitting of characters means that they are given greater weight in the analysis, but equally the lumping of characters leads to underweighting. Our sensitivity analyses indicate that our current character set does not bias our analysis towards a certain topology. Consequently, we argue that the best protocols for dealing with such issues are to check for co-variation in scores, test for correlated characters (e.g., Maddison, 2000; Maddison & FitzJohn, 2015; Randle & Sansom, 2016), and to consider weighting methods (e.g., Wiens, 2001; Goloboff, Mattoni & Quinteros, 2006), rather than to exclude potentially phylogenetically informative variation. If state scores are incongruent between two characters for a set of taxa, then this is evidence that they are not correlated and thus they should be treated as separate characters. Finally, the data matrix of Mannion et al. (2013) has already undergone several rounds of character revision and addition (Poropat et al., 2015a; Poropat et al., 2015b; Poropat et al., 2016; Upchurch, Mannion & Taylor, 2015; this study), which will continue with future iterations. Here we have focused on characters relevant to basal Macronaria, but future work will examine potentially problematic characters that are pertinent to other parts of the tree (e.g., Titanosauria).

Phylogenetic inter-relationships of Brachiosauridae

Comparisons with the topologies of two recent independent analyses (Carballido et al., 2015; D’Emic, Foreman & Jud, 2016) reveal a number of similarities, as well as some differences in brachiosaurid inter-relationships. Whereas Carballido et al. (2015) recovered Padillasaurus as a brachiosaurid, our study places it in Somphospondyli (see below; note that this taxon was described after D’Emic, Foreman & Jud (2016) was accepted for publication). All three analyses place Europasaurus as basal to the other brachiosaurids, although in Carballido et al. (2015) Europasaurus lies outside of Titanosauriformes (see below). Our analysis is the only one to include Vouivria, and so comparisons cannot be made for the position of this taxon. Brachiosaurus occupies a basal position in our analysis, but it is nested with more derived brachiosaurids in the other two studies, forming a clade with Abydosaurus + Giraffatitan in Carballido et al. (2015), whereas it clusters with Abydosaurus, Lusotitan, Cedarosaurus and Venenosaurus in D’Emic, Foreman & Jud (2016). Although its position is poorly resolved, our analysis suggests that Abydosaurus belongs to this clade of more derived brachiosaurids and is not closely related to Brachiosaurus. Sonorasaurus was not included in the analysis of Carballido et al. (2015), but is recovered as the sister taxon to Giraffatitan here and in D’Emic, Foreman & Jud (2016). Our analysis places Lusotitan in a similar position as that of D’Emic, Foreman & Jud (2016), whereas this taxon falls out in an unresolved basal titanosauriform position in Carballido et al. (2015). All three analyses recover Cedarosaurus and Venenosaurus as closely related or sister taxa.

Nearly all of these brachiosaurid taxa have been described in detail. Only the appendicular skeleton of Europasaurus remains to be described of this taxon, but our OTU here is based on first-hand study of these materials by PDM. Abydosaurus, the most unstable brachiosaurid OTU in our analysis, is the only taxon for which a detailed description is currently lacking (Chure et al., 2010), and we have not been able to study this material. As such, a future iteration of this data matrix that incorporates all anatomical information for Abydosaurus might enable the full resolution of brachiosaurid inter-relationships, but novel information is otherwise likely to only come from the discovery and description of new specimens.

Following our revised analysis, Brachiosauridae is supported by the following synapomorphies: (1) presence of denticles on teeth (C113 (reversal)); (2) lateral fossa at the base of the prezygapophyseal process in middle cervical vertebrae (C403); (3) elongate and dorsoventrally narrow diapophyses in anterior–middle dorsal vertebrae (C154); (4) anterior dorsal neural spines that are dorsoventrally taller than posterior ones (C158); (5) middle–posterior dorsal neural spines with spinopostzygapophyseal laminae divided into medial and lateral branches (C165); (6) ribs of sacral vertebra 2 include a contribution from Sv1 (C406; Europasaurus = ‘?’); (7) ventral margin of coracoid without a notch anterior to glenoid (C220 (reversal)); (8) anteroposterior length of proximal plate to proximodistal length of ischium ratio of 0.25 or less (C61); (9) small ischial contribution to the acetabulum (C62); (10) maximum mediolateral width to dorsoventral height ratio of the proximal end of metacarpal III is 1.3 or greater (C411; Europasaurus = ‘?’) (see also Wilson, 2002; Taylor, 2009; D’Emic, 2012; Mannion et al., 2013; D’Emic, Foreman & Jud, 2016; this study).

Europasaurus is a brachiosaurid

Consistent with previous iterations of this data matrix, the Late Jurassic German dwarf sauropod Europasaurus is recovered as a brachiosaurid, and is placed as the most basal member of this clade (see also D’Emic, 2012; D’Emic, 2013; D’Emic, Foreman & Jud, 2016). However, this position differs from other sauropod analyses in which Europasaurus has been recovered as a non-titanosauriform macronarian (Sander et al., 2006; Ksepka & Norell, 2010; Carballido et al., 2011a; Carballido et al., 2011b; Carballido et al., 2015; Carballido & Sander, 2014; Royo-Torres et al., 2014).

Our Europasaurus OTU includes several revised character scores following Carballido & Sander (2014) and Marpmann et al. (2015) (note that some of these were revised by Poropat et al., 2016). Two of these changes correspond to features that originally supported brachiosaurid affinities, but have been demonstrated to be absent or incorrectly scored for Europasaurus: (1) the distance separating the supratemporal fenestrae relative to the long axis of each supratemporal fenestra (C4) varies with ontogeny, and more mature specimens of Europasaurus demonstrate this ratio to be greater than 1.0 (Marpmann et al., 2015), in contrast to Abydosaurus and Giraffatitan (as well as several other eusauropods); and (2) maxillary teeth are not strongly twisted (C114), with this feature instead restricted to Abydosaurus and Giraffatitan (Marpmann et al., 2015).

At least three of our new/revised scores result in Europasaurus being characterised by features that are unusually plesiomorphic for a placement within Titanosauriformes: (1) Europasaurus has an undivided cranial nerve II (C398), whereas this is divided in nearly all neosauropods (Sander et al., 2006; Marpmann et al., 2015; this study); (2) the absence of camellae in the cervical–anterior dorsal vertebrae (C115) of Europasaurus (Carballido & Sander, 2014) contrasts with that of Galveosaurus + Titanosauriformes (Mannion et al., 2013; although it is possible that Vouivria also lacks camellae—see above); and (3) the absence of camerae in the middle–posterior dorsal vertebrae (C141) of Europasaurus (which are essentially ‘solid’; Carballido & Sander, 2014) contrasts with eusauropods more derived than Omeisaurus (Wilson & Sereno, 1998). It is possible that such features (especially C141 and C398) reflect the paedomorphic retention of plesiomorphic characters in Europasaurus, although they seem to have no effect on our topology, which is unchanged when these three characters are instead each scored as missing data.

We are unsure why this difference in the position of Europasaurus (and Lusotitan) remains between our analysis (and D’Emic, Foreman & Jud, 2016) and that of iterations of the Carballido et al. (2015) matrix. One possibility pertains to taxon sampling. In particular, Haplocanthosaurus priscus and Bellusaurus are recovered at the base of Macronaria in the Carballido et al. (2015) matrix, but neither taxon is included in the matrix of D’Emic, Foreman & Jud (2016), or that used in the present study. It is possible that one or both of these taxa impacts upon character polarity such that the composition of Titanosauriformes is affected. However, when we re-ran the Carballido et al. (2015) matrix with these two taxa excluded a priori (otherwise using the same analytical protocols), the topology of Macronaria was unchanged (25 MPTs of 1,056 steps). To investigate the effect of taxon sampling further, we reduced the Carballido et al. (2015) matrix to only those taxa included in our study (38 taxa included, with 35 taxa excluded). Aside from some loss of resolution, re-running the analysis (with Shunosaurus set as the outgroup) resulted in an essentially unchanged topology (720 MPTs of 765 steps), with Europasaurus still placed outside of Titanosauriformes. As such, taxon sampling is not the cause of the discrepancy between the matrices, and the differences must pertain to variation in character scoring and/or character sampling. Finally, we note that two taxa (Euhelopus and Tastavinsaurus) that are otherwise universally recovered within Titanosauriformes also fall out as non-titanosauriform macronarians in the Carballido et al. (2015) matrix (including our subsampled datasets).

Padillasaurus is a somphospondylan

As noted by Carballido et al. (2015), the main anatomical feature driving a brachiosaurid placement for the Early Cretaceous Colombian sauropod Padillasaurus is the presence of a ‘blind’ fossa in its anterior caudal centra. This feature has otherwise only been documented in Abydosaurus, Cedarosaurus, Giraffatitan and Venenosaurus, and has been recovered as a synapomorphy of a subset of brachiosaurids in several analyses (D’Emic, 2012; Mannion et al., 2013; Carballido et al., 2015; D’Emic, Foreman & Jud, 2016). However, the recent discovery of the Australian titanosaur Savannasaurus has demonstrated the presence of this blind fossa in a non-brachiosaurid titanosauriform (Poropat et al., 2016). Although the exclusion of Savannasaurus a priori does not result in Padillasaurus being united with Brachiosauridae, it does greatly increase the instability of this taxon, leaving it in an unresolved polytomy within basal titanosauriforms. In summary, our increased taxon sampling demonstrates the wider distribution of the blind fossa in sauropod caudal vertebrae, and argues for somphospondylan, rather than brachiosaurid, affinities for Padillasaurus. This placement is more in keeping with our current knowledge of Cretaceous South American titanosauriforms, all of which appear to be somphospondylans (Mannion et al., 2013; Jesus Faria et al., 2015).

Evolutionary history of Brachiosauridae

Purported Middle Jurassic brachiosaurids have all been demonstrated to belong outside of Neosauropoda (D’Emic, 2012; Mannion et al., 2013), with anatomical similarities such as forelimb elongation (e.g., Atlasaurus) and gracility (e.g., Lapparentosaurus) reinterpreted as convergence. Currently, Late Jurassic members of Brachiosauridae are only definitely known from East Africa (Janensch, 1914), western Europe (Mannion et al., 2013; Mocho, Royo-Torres & Ortega, 2016; this study), and the USA (Riggs, 1903a). A Late Jurassic South American macronarian specimen (Rauhut, 2006) cannot confidently be assigned to the clade (Mannion et al., 2013).

Brachiosaurid remains are known from the earliest Cretaceous of southern Africa (McPhee et al., 2016), and possibly from Afro-Arabia too (Buffetaut et al., 2006). Several taxa (Abydosaurus, Cedarosaurus, Sonorasaurus and Venenosaurus) and additional indeterminate remains (see Taylor, 2009) from the late Early Cretaceous of the USA are referable to Brachiosauridae (Chure et al., 2010; D’Emic, 2012; D’Emic, 2013; D’Emic, Foreman & Jud, 2016). The contemporaneous North American sauropod Sauroposeidon (Wedel, Cifelli & Sanders, 2000; including material originally named Paluxysaurus (Rose, 2007)) was originally regarded as a brachiosaurid, but its somphospondylan affinities have been subsequently demonstrated in several studies (D’Emic, 2012; D’Emic, 2013; Mannion et al., 2013; Carballido et al., 2015). No Cretaceous sauropod remains from Europe or South America can be referred to Brachiosauridae (Mannion et al., 2013; Royo-Torres et al., 2014; this study).

There is currently no evidence that brachiosaurids ever reached Asia (Ksepka & Norell, 2010; Mannion, 2011), despite good sampling of suitable deposits (Upchurch, Barrett & Dodson, 2004). Brachiosaurid remains have also not been discovered in Australasia or Antarctica, although we can have less confidence that their absence from these landmasses is a true signal, rather than an artefact of patchy sampling (Mannion et al., 2013).

Vouivria damparisensis represents the stratigraphically oldest member of Brachiosauridae, and the clade spanned the middle–late Oxfordian (early Late Jurassic) through to the late Albian/early Cenomanian (mid-Cretaceous), with the last known occurrences all from North America (D’Emic, Foreman & Jud, 2016) (Fig. 40). Regardless of whether their absence from the Cretaceous of Europe, as well as other regions entirely, reflects regional extinctions and genuine absences, respectively, or sampling artefacts, brachiosaurids appear to have become globally extinct by the earliest Late Cretaceous.

Conclusions

A detailed redescription of a long-neglected sauropod specimen from the middle–late Oxfordian (Late Jurassic) of eastern France recognises it as a distinct brachiosaurid, Vouivria damparisensis n. gen. n. sp., that is the stratigraphically oldest known occurrence of Titanosauriformes. An expanded and revised phylogenetic analysis includes the dwarfed Late Jurassic European taxon Europasaurus within Brachiosauridae, but places the Early Cretaceous Colombian genus Padillasaurus within Somphospondyli. The past distribution of Brachiosauridae is currently restricted to Europe, North America and Africa, and the clade appears to have become globally extinct by the earliest Late Cretaceous.

Supplemental Information

Supplemental Information 1 Supplementary Information

Click here for additional data file.

Fig. S1 Palaeogeography of the carbonated Jura platform during the upper Oxfordian–lower Kimmeridgian

Map modified after Cariou (2013). Abbreviated locations for: (1) truncations: O, Ornans (Enay, Contini & Boullier, 1988), BP, Bonnevaux-le-Prieuré (Lathuiliere et al., 2005), Damparis (this study); and (2) fossil woods: BC, Besançon (Citadelle) (Bulle et al., 1968): characean algae horizons; BG, Besançon (Brégille) (Contini, 1972): basal black marls with Brachyphyllum moreauanum, imprints of Zamites and Zonarites gracillimus; CS, Charbonnière-les-Sapins (Merle, 1905): lignite horizon with characean algea separating the Rauracian from the Sequanian, with Zamites and characean pinnules; D, Dambelin (Enay, Contini & Boullier, 1988): grey-black marls; Gi, Gilley (Oertli & Ziegler, 1958): gyrogonites of characean algae and shells of limnic ostracods from freshwater ponds; Gr, Grattery (Contini, 1972) 10 m above the base: vegetation imprints: fronds and pinnules of Zamites feneonis and Zamites formosus, pieces of leaf stems of Brachyphyllum moreauanum; H, Hyémondans (Enay, Contini & Boullier, 1988): basal black marls with wood debris; RV, Roche-sur-Vannon (Glangeaud, 1944): Zamites feneonis, Brachyphyllum moreauanum, Cycadospermum (fruits), young pinecones of evergreens, algae.

Click here for additional data file.

Table S1 Geological correlation scheme

Correlation scheme of the lithological units of Damparis with both synthetic geological successions of Franche-Comté region and of north-western Switzerland.

Click here for additional data file.

Table S2 Biostratigraphical distinction of the geological formations from the late Middle and Upper Oxfordian in Franche-Comté (France)

Taxon names and stratigraphical ranges have been updated from original publications and different sources respectively. Only two taxa (highlighted in yellow) are still affected by taxonomic vs. stratigraphic inconsistencies.

Click here for additional data file.

Data S1 Data matrix (TNT and Mesquite)

Click here for additional data file.

The authors would like to thank both the Solvay group and Inovyn company for allowing fieldwork in the Damparis quarry, François Atrops and Raymond Enay from Lyon 1 University for the identification of the ammonites, and Elsa Cariou from Nantes University for stimulating discussion. We thank Y Després and V Pernègre for help in the cleaning and re-preparation of the specimens. Specimen photographs were taken by L Cazes of the MNHN, with collaboration of RA. We are also grateful to all of those who have helped with access to sauropod specimens in their care. José Carballido kindly provided us with additional information and photographs of Padillasaurus. Paul Upchurch also provided helpful discussion on interrelated characters. Reviews by Emanuel Tschopp and Mathew Wedel improved an earlier version of this work.

Institutional Abbreviations

BYU Brigham Young University, Provo, Utah, USA

CPT Museo de la Fundacíon Conjunto Paleontológico de Teruel-Dinópolis, Aragón, Spain

MCF Museo ‘Carmen Funes’, Neuquén, Argentina

MfN Museum für Naturkunde Berlin, Germany

ML Museu da Lourinhã, Lourinhã, Portugal

MCNV Museo de Ciencias Naturales de Valencia, Spain

MNHN Muséum National d’Histoire Naturelle, Paris, France

MNN Musée National du Niger, Niamey, Republic of Niger

PVL Colección de Paleontología de Vertebrados de la Fundación Instituto Miguel Lillo, Tucumán, Argentina

SMNS Staatliches Museum für Naturkunde Stuttgart, Germany

Additional Information and Declarations

Competing Interests

Author Contributions

Data Availability

New Species Registration

The authors declare there are no competing interests.

Philip D. Mannion conceived and designed the experiments, performed the experiments, analyzed the data, contributed reagents/materials/analysis tools, wrote the paper, prepared figures and/or tables, reviewed drafts of the paper.

Ronan Allain and Olivier Moine conceived and designed the experiments, contributed reagents/materials/analysis tools, wrote the paper, prepared figures and/or tables, reviewed drafts of the paper.

The following information was supplied regarding data availability:

The raw data has been supplied as a Supplementary File.

The following information was supplied regarding the registration of a newly described species:

Vouivria: urn:lsid:zoobank.org:act:B06BCF72-56A8-4DD6-BC27-CC0BA6D0092D, damparisensis: urn:lsid:zoobank.org:act:CCAA960C-6A39-46A4-8AC9-70D8BA816647, Publication LSI: urn:lsid:zoobank.org:pub:5EB3AF68-A8A2-407D-BF1B-5F856C3B505D.

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
