# Peer review of "The earliest known titanosauriform sauropod dinosaur and the evolution of Brachiosauridae"

_PeerJ, doi:10.7717/peerj.3217_

## Round 0.1 · original submission · Minor Revisions

Thank you for your submission of this important work and sorry for the delay in issuing the first decision on this rather long manuscript. I am pleased to report that the two reviewers praised the quality and completeness of your work. Each of them provided a detailed report and made some suggestions to clarify and/or complete different parts of the manuscript. Please, review these and revise your manuscript accordingly.

Among the most important points raised by the reviewers are:

- The need for a clarification of how much of the geological work has been done in the context of your study, and how much is derived from previous studies. Linked to that is the fact that the Supp. Info. should probably be more tightly and more clearly integrated in the Geological part of your manuscript.

- Some points raised by the second reviewer regarding the phylogenetic analysis and the discussion of its results.

·

Basic reporting

The current manuscript is a clear successor to Mannion et al. (2013), but it embodies several important advances: (1) much more detailed geological and morphological descriptive work on what will now be Vouivria (formerly the French 'Bothriospondylus'), (2) a revised and expanded phylogenetic analysis focusing on the Brachiosauridae, and (3) an interesting and useful discussion of character correlation in sauropod phylogenetic work, nicely illustrated with examples from this very work. The rationale and organization of the paper is clear (for the most part - see comments below), and the comparative work and citations of the literature are admirably comprehensive and up to date. My suggestions for improvement are few.

The only section of the manuscript that is less than clear is 'Geographical and Geological Context' (lines 131-266). The first and most major problem here is that it is often unclear how much of the descriptive work is new and being presented by these authors for the first time, and how much is derived from previous studies. A simple, straightforward declaration at the start of this section is badly needed, along the lines of, "We revisited the Damparis quarry to remeasure the stratigraphic section, take samples, etc." Also, although Figure 6 is helpful, the manuscript would benefit considerably from the addition of a standard stratigraphic column for the Damparis quarry.

The second problem in this section is that material that is more fully explained in the Supplementary Information is mentioned here without any explanatory context or reference to the SI. For example, the sentence in lines 246-248, "Locally it represents the Ox5 third order cycle sequence boundary..." is fairly clear once the reader has digested the SI, but almost completely unparseable without having read the SI - and neither that sentence nor the surrounding ones direct the reader to the SI to resolve the confusion. This needs to be remedied by a clear statement of what material is in the body of the paper versus the SI, and by explicit reference to the SI when otherwise unfamiliar terms are introduced (e.g., 'Ox5 third order cycle sequence boundary').

Third, the writing throughout the geology section is less clear than in the rest of the paper. Two examples will suffice. In the short paragraph in lines 214 to 217, the tense switches from past to present and then back to past again. Combined with the work attribution problem mentioned above, this makes it very difficult to determine whether the authors are describing the current or past condition of the quarry, based on their work or the work of others. And it seems very unlikely that anyone will understand the sentence in lines 320-323 on the first pass - I had to read it about four times to make sure I was following the logic. I recommend to the authors the old trick of reading the manuscript aloud, as a mechanism for discovering and fixing the occasionally awkward constructions.

Finally for the geology section, Figure 6 would be a lot more helpful if the features in the photographs were numbered from top to bottom or vice versa, instead of scrambled as they are in the current draft.

Experimental design

Nothing to add.

Validity of the findings

Nothing to add.

Additional comments

Minor comments

Lines 685-686, "Although we agree that it is more parsimonious to assume that different individuals of the same species underwent the same sequence of sacral fusion..." According to Table 1 in Wedel and Taylor (2013) - already cited herein - postcranial fusion sequences in sauropods vary widely even within genera. More grist for the mill, if any is needed.

Lines 733-735, "The ventral surface is transversely convex, curving smoothly into the lateral surfaces, without forming a distinct ventral surface." This needs to be worded more clearly, without implying that the vertebra lacks the very feature being described (i.e., a ventral surface).

Lines 1088-1089, "The medial surface of the ilium preserves a number of muscle scars for sacral rib attachments." Surely these must be articular surfaces rather than muscle scars?

Lines 1514-1517, the words "it might not seem entirely unreasonable to consider that" can be omitted from the sentence with no loss of meaning, with an improvement in force and clarity, and with a decrease in peevishness - even if the last was unintentional.

·

Basic reporting

Dear Jérémy Anquetin,

many thanks for inviting me to review this important article. This is a very detailed and extensively figured description of a crucial specimen to understand sauropod evolution. Mannion et al. also put the find in a highly detailed geological context, and the study will therefore also be of importance for researchers interested in Jurassic stratigraphy in general.

There are just a few methodological points I'd like to raise, which however could possibly result in extensive changes in the interpretation of the results.

I therefore suggest acception of the article, but have difficulties to decide if after minor or major revisions. I suppose for the workload, the proposed changes would rather be major, but they are few in number, and I don't feel they would have to be checked again, so according to your guidelines this will technically be minor changes.

The description and geological context are done in great detail, and the figures are generally accurate and helpful (more detailed comments on this below). Data on the phylogenetic analysis is shared sufficiently so that independent test could be made.

One small point: comparisons of the manus and pes material with other taxa might profit from my recent paper on the manus and pes of Camarasaurus, where I also made quite some comparisons with brachiosaurid taxa (Tschopp et al. 2015b).

Experimental design

The study is generally well designed, with clear, relevant questions and results fitting the scope of the journal.

I don't have the knowledge and expertise to evaluate the geological and stratigraphical analyses, so I won't comment on these here. I hope another referee will be more competent here.
I'll instead highlight a few issues with the phylogenetic analysis, that I found potentially questionable, and which should at least be addressed in the text, in my opinion.

All the data was provided, so that I could run an independent test. I ran the matrix under implied weights (k=5), and new technology search with all search algorithms enabled and the consensus stabilized five times (being a rather preliminary analysis, I didn't run a second round of TBR. In my experience such a second round rarely finds shorter trees, and also rarely changes the topology in the strict consensus tree, so I feel confident in sharing the results). The resulting strict consensus tree is slightly different from the one presented in the paper (which is based on equal weighting, as far as I understood, but this is not explicitly stated in the MS. Please state it in the revised version).
1) the 9 “unstable” taxa pruned a priori could actually be found in relatively defined positions along the tree, which might argue against an a priori exclusion
2) Vouivria is found as the sister taxon to Lusotitan, which might make more sense in a biogeographic point of view than the topology proposed in the MS
3) TNT recovers 5 autapomorphies for Vouivria (26-0, 225-1, 268-0, 369-1, 407-1; character number already corrected), but three of them are shared with other brachiosaurid taxa
4) all of the three uniting synapomorphies of Vouivria + Lusotitan (see attached files) are shared with at least one of the more basal taxa Jobaria, Lapparentosaurus, and Atlasaurus. Some of these, in earlier analyses, were found together with members of Turiasauria, which is a clade that is generally regarded as having a number of convergent features with brachiosaurids. Also in the present analysis, these three taxa are recovered in the position where the turiasaur clade is generally found in phylogenetic analyses that also include more basal eusauropods (e.g. Carballido et al. 2012, 2015; Royo-Torres et al. 2014). However, the current analysis does not include any of the best known members of this clade (e.g. Turiasaurus riodevensis or Losillasaurus giganteus), and therefore excludes the recognition of this clade a priori. Therefore, it also hampers a potential identification of Vouivria as turiasaur.

Because of this, I'd like to see some discussion of these points in the final paper. This does not necessarily have to be the inclusion of more turiasaurs in the matrix (although I would personally prefer that), but the possibility should at least be discussed in more detail. Also, I suggest to add an analysis under implied weights, especially because of the better fit with biogeography.

Validity of the findings

The description is detailed and accurate, and the phylogenetic analysis is based on well-established data. However, validity of the results depends on the discussion of the doubts raised above after my independent phylogenetic analysis, especially concerning biogeography, and systematic placement.

Additional comments

Introduction
Would it be possible to add a figure showing the several interpretations of how Brachiosauridae is constituted in the various analyses discussed in the Introduction (lines 112-122)?
Diagnosis
Please figure the diagnostic features, or indicate them in the photographs.
The flange on mtc III (autapomorphy 5) might actually also be present in Camarasaurus (see Tschopp et al. 2015), and the bifurcation in Jobaria (based on figures I obtained from José Carballido).
Description and Comparison
It would be nice to have some more information on bone terminology, and how you apply orientational terms to bones that are oriented obliquely in the skeleton (like the scapula or metapodials).
Please also provide some more detailed information on how you identified the serial positions of the vertebrae.
Teeth
Lines 455-456: had to read them twice to understand what you mean, please consider rephrasing
Can't you identify the orientation of the teeth? In my experience, denticles (if only present on one side) are generally on the mesial edge (e.g. USNM 5730), but I haven't studied this in detail so this might be misleading.
Line 470: Does the “anastomosing wrinkles” pattern follow the terminology proposed by Holwerda et al. 2015?
Line 478: why is it likely to be taphonomic?
Line 484: couldn't it be that the denticles result in slower wear on the side they are preserved? If so, this pattern could also be explained as being a result of initial wear, producing a wear facet on the side without denticles, and having ground down the denticles on the other side, but not yet to the degree of producing a wear facet.
Cervical vertebrae
Line 538: where in the fossae are the small foramina located?
Line 566: such a distinct posterior expansion of the SPOL at the spine summit also occurs in anterior cervicals of Galeamopus SMA 0011 and in certain camarasaur caudals
Dorsal vertebrae
Lines 627-630: This morphology is also present in the last dorsal vertebrae of the FMNH Brachiosaurus holotype specimen (I can send you pictures if you need; by the way, this is what I described as “slight opisthocoely” in my diplodocid phylogeny, based on Carballido et al. 2012)
Thoracic ribs
Is is possible to provide some figures?
Sacral vertebrae
Line 704: delete “too”
Is there any transverse ridge on the ventral surface of any sacral rib, as occurs in some apatosaurs and camarasaurs (see Mook 1917; Ostrom & McIntosh 1966).
Caudal vertebrae
Line 738: caudal pneumaticity in rebbachisaurs is also described in Fanti et al. 2013, 2015
Scapula
an anterior ventral process is also present in some diplodocoids (e.g. Supersaurus, Diplodocus; Tschopp et al. 2015a)
Coracoid
The fact that the medial surface is predominantly flat seems peculiar to me. This is very different from what you see in Camarasaurus. How much of this could be due to deformation?
Manus
please provide some more information on how you identified the manual phalanges as manual and based on which features you attributed them to specific digits
Ilium
Can we infer if the articular surface of the pubic peduncle was significantly wider than long, even if it's not preserved completely?
Ischium
Line 1135: “dorsoventral height” of the blade: what do you mean with this exactly? The blade often twists along its axis, or is oriented obliquely in life. Please explain better what you're actually measuring here.
Femur
Line 1170: “intercondylar ridges”. Can you indicate these on the figures? I don't think I have ever seen anything similar. Is this also present in other taxa?
Tibia
Line 1181: add “to” after “A poorly preserved fragment appears”
Line 1194: can you indicate the “small pointed projection on the figures?
Dataset
Line 1331: “three”, not “two” new characters
Added characters
Can you provide the character numbers in the other analyses, where you took them from? Also, can you illustrate these characters, or refer to already published figures?
C404: it might be worth to define the location of the coels within the fossa more clearly. Otherwise the three coels in Supersaurus lourinhanensis would have to be scored equally as brachiosaurs
Character correlation
While I generally agree with the fact that we should include as many characters as possible, I fear that your argument is somewhat weak in some cases. For instance, concerning the correlated forelimb gracility characters, the differential scoring pattern in the matrix actually depends at least in parts on how you define the state boundaries: what you define as “gracile” in the humerus might not correspond to how you define it in the ulna and radius, but their gracility could still be correlated.
Line 1484: ratios of the articular surfaces of vertebral centra seem to be quite easily affected by deformation (see Tschopp et al. 2013), so you should be cautious anyway in using these characters
Lines 1524-1525: this sentence is confusing: 33 taxa could be scored for one or both, and 20 for both characters? So the 20 are part of the 33? Please consider rephrasing to reduce confusion.
Europasaurus is a brachiosaurid
Maybe the difference in topology is due to different taxon sampling? Carballido has much more basal OTUs. Also, you don't have Haplocanthosaurus in your matrix here, which might have some influence on basal macronarian interrelationships.
Tables
please provide some more detailed measurement protocols (especially for neural arch height and neural spine height; table 3)
Figure 37
Indicate also in the caption that you excluded 9 taxa a priori. The provided TNT and Mesquite files still have them.
References:
Carballido, J. L., D. Pol, M. L. P. Ruge, S. P. Bernal, M. E. Páramo-Fonseca, and F. Etayo-Serna. 2015. A new Early Cretaceous brachiosaurid (Dinosauria, Neosauropoda) from northwestern Gondwana (Villa de Leiva, Colombia). Journal of Vertebrate Paleontology 35:e980505.
Carballido, J. L., L. Salgado, D. Pol, J. I. Canudo, and A. Garrido. 2012. A new basal rebbachisaurid (Sauropoda, Diplodocoidea) from the Early Cretaceous of the Neuquén Basin; evolution and biogeography of the group. Historical Biology 24:631–654.
Fanti, F., A. Cau, M. Hassine, and M. Contessi. 2013. A new sauropod dinosaur from the Early Cretaceous of Tunisia with extreme avian-like pneumatization. Nature Communications 4.
Fanti, F., A. Cau, L. Cantelli, M. Hassine, and M. Auditore. 2015. New information on Tataouinea hannibalis from the Early Cretaceous of Tunisia and implications for the tempo and mode of rebbachisaurid sauropod evolution. PLOS ONE 10:e0123475.
Holwerda, F. M., D. Pol, and O. W. M. Rauhut. 2015. Using dental enamel wrinkling to define sauropod tooth morphotypes from the Cañadón Asfalto Formation, Patagonia, Argentina. PLOS ONE 10:e0118100.
Mook, C. C. 1917. Criteria for the determination of species in the Sauropoda, with description of a new species of Apatosaurus. Bulletin of the American Museum of Natural History 37:355–358.
Ostrom, J. O., and J. S. McIntosh. 1966. Marsh’s Dinosaurs: The Collection from Como Bluff. Vol. 1. Yale University Press, New Haven, pp.
Royo-Torres, R., P. Upchurch, P. D. Mannion, R. Mas, A. Cobos, F. Gascó, L. Alcalá, and J. L. Sanz. 2014. The anatomy, phylogenetic relationships, and stratigraphic position of the Tithonian–Berriasian Spanish sauropod dinosaur Aragosaurus ischiaticus. Zoological Journal of the Linnean Society 171:623–655.
Tschopp, E., J. Russo, and G. Dzemski. 2013. Retrodeformation as a test for the validity of phylogenetic characters: an example from diplodocid sauropod vertebrae. Palaeontologia Electronica 16:1–23.
Tschopp, E., O. Mateus, and R. B. J. Benson. 2015a. A specimen-level phylogenetic analysis and taxonomic revision of Diplodocidae (Dinosauria, Sauropoda). PeerJ 3:e857.
Tschopp, E., O. Wings, T. Frauenfelder, and W. Brinkmann. 2015b. Articulated bone sets of manus and pedes of Camarasaurus (Sauropoda, Dinosauria). Palaeontologia Electronica 18:1–65.

---

## Round 0.2 · accepted · Accept

Thank you for this thorough revision of your manuscript. You complied with and/or responded to all of the comments made by the two reviewers and I am now in a position to accept your submission for publication in PeerJ. Congratulations on this comprehensive work.